# Molecular insights into the catalytic promiscuity of a bacterial diterpene synthase

Zhong Li[1,6], Lilan Zhang[2,6], Kangwei Xu [3,6], Yuanyuan Jiang[1], Jieke Du[1], Xingwang Zhang [1], Ling-Hong Meng[4,5], Qile Wu[1], Lei Du[1], Xiaoju Li[1], Yuechan Hu[2], Zhenzhen Xie[2], Xukai Jiang[1], Ya-Jie Tang[1], Ruibo Wu [3] ✉, Rey-Ting Guo [2] ✉ & Shengying Li [1,5] ✉

Diterpene synthase VenA is responsible for assembling venezuelaene A with a unique 5-5-6-7 tetracyclic skeleton from geranylgeranyl pyrophosphate. VenA also demonstrates substrate promiscuity by accepting geranyl pyrophosphate and farnesyl pyrophosphate as alternative substrates. Herein, we report the crystal structures of VenA in both *apo* form and *holo* form in complex with a trinuclear magnesium cluster and pyrophosphate group. Functional and structural investigations on the atypical [115]DSFVSD[120] motif of VenA, *versus* the canonical Asp-rich motif of DDXX(X)D/E, reveal that the absent second Asp of canonical motif is functionally replaced by Ser116 and Gln83, together with bioinformatics analysis identifying a hidden subclass of type I microbial terpene synthases. Further structural analysis, multiscale computational simulations, and structure-directed mutagenesis provide significant mechanistic insights into the substrate selectivity and catalytic promiscuity of VenA. Finally, VenA is semi-rationally engineered into a sesterterpene synthase to recognize the larger substrate geranylfarnesyl pyrophosphate.

Terpenoids (terpenes and modified terpenes with heteroatoms) represent the largest family of natural products with over 100,000 members[1]. These important primary and secondary metabolites, which are widely distributed in plants, animals, and microbes, hold diverse physiological and ecological functions[2–5]. Biosynthetically, the diversity of polycyclic and chiral hydrocarbon skeletons derive from various cyclization modes of a small number of acyclic and achiral polyprenyl pyrophosphates, which are catalyzed by highly diversified terpene synthases (TPSs)[6]. These elegant transformations are initiated by a highly reactive carbocation, followed by a cascade of intramolecular olefin-to-cation cyclizations, hydride shifts, proton transfer, Wagner–Meerwein rearrangements, and terminal deprotonation[7,8].

Typical TPSs are divided into two main classes based on sequence characteristics and initial carbocation forming mechanisms[9–12]: the type I TPSs containing two conserved motifs including the Asp-rich motif DDXX(X)D/E and NSE/DTE motif with a consensus sequence of (N/D)DXX(S/T)XXXE, which co-mediate the ionization of substrate by pyrophosphate (PP$_i$) abstraction; and the type II TSs possessing an Asp-rich DXDD motif to protonate the olefin or epoxy group of the substrate.

Approximately one-quarter of discovered terpenoids are diterpenoids (C$_{20}$), many of which have been successfully applied as medicines, perfumes, food additives, and fine chemicals, and also provided abundant target molecules for both synthetic chemistry and synthetic

[1]State Key Laboratory of Microbial Technology, Shandong University, No. 72 Binhai Road, Qingdao, Shandong 266237, China. [2]State Key Laboratory of Biocatalysis and Enzyme Engineering, Hubei Hongshan Laboratory, Hubei Collaborative Innovation Center for Green Transformation of Bio-Resources, Hubei Key Laboratory of Industrial Biotechnology, School of Life Sciences, Hubei University, Wuhan, Hubei 430062, China. [3]School of Pharmaceutical Sciences, Sun Yat-sen University, Guangzhou, Guangdong 510006, China. [4]Key Laboratory of Experimental Marine Biology, Institute of Oceanology, Chinese Academy of Sciences, Nanhai Road 7, Qingdao, Shandong 266071, China. [5]Laboratory for Marine Biology and Biotechnology, Qingdao National Laboratory for Marine Science and Technology, Qingdao, Shandong 266237, China. [6]These authors contributed equally: Zhong Li, Lilan Zhang, Kangwei Xu. ✉e-mail: wurb3@mail.sysu.edu.cn; guoreyting@hubu.edu.cn; lishengying@sdu.edu.cn

biology[1,13]. For example, paclitaxel from *Pacific yew* is a first-line anti-tumor agent clinically used in cancer therapies[14]; vitamin A, an essential micronutrient of humans, plays important roles in growth, development, vision, and reproduction[15]; gibberellins, mainly produced by plants (and some microbes), are vital plant hormones for regulation of growth and development[16]; ambroxide, as a commercial perfume fragrance, is semi-synthesized from sclareol that is a diterpenoid natural product isolated from *Salvia sclarea*[17]. To date, over 100 carbon skeleton types (e.g., labdane-, beyerene-, kaurene-type, etc.) of diterpenes have been discovered, demonstrating fascinating structural and functional diversities and many interesting but unknown catalytic mechanisms of diterpene synthases (DTSs). Thus, it is a long-term goal for biochemists, structural biologists, computational chemists, and synthetic chemists/biologists to understand how versatile DTSs assemble these diverse, complex, and useful skeletons (Fig. 1a, b)[18–21].

Plant-derived TPSs used to catch much attention since over 70% of terpenoids originated from plants[1,22,23]. With the explosively increased genomic data, bacteria have been found to be another important source of TPSs and terpenoids[24,25]. Recently, we discovered a fragrant and low-toxic diterpene venezuelaene A (**1**) with an unusual 5-5-6-7 tetracyclic skeleton (Fig. 1c) from the model actinomycetic strain *Streptomyces venezuelae* ATCC 15439 by genome mining and silent gene activation[26], which has been attracting growing interests from both perfume industry and (bio)chemists due to its *trans*-fused bicyclo[3.3.0]octane ring system[27,28]. A promiscuous DTS VenA containing an atypical $^{115}$DSFVSD$^{120}$ motif was characterized to catalyze the unique cyclization of geranylgeranyl pyrophosphate (GGPP, $C_{20}$), giving rise to **1**. VenA is also capable of converting farnesyl pyrophosphate (FPP, $C_{15}$) into hedycaryol (**2**), germacrene A (**3**), $\beta$-farnesene (**4**), and germacrene D (**5**), and of accepting geranyl pyrophosphate (GPP, $C_{10}$) to form geraniol (**6**) with relatively lower efficiencies (Fig. 1c). However, VenA cannot use geranylfarnesyl pyrophosphate (GFPP, $C_{25}$) as a substrate. The mechanism for **1** biogenesis was proposed to be initiated by Markovnikov 1,10-cyclization based on isotopic labeling experiments[26]. However, the structural basis and molecular

**a   1,11-Cyclization of GGPP**

Cyclooctat-9-en-7-ol (5-8-5)

Spiroviolene (5-5-5-5)

Fusicocca-2,10(14)-diene (5-8-5)

Cattleyene (5-6-5-5)

CotB2
$^{110}$DDMD$^{113}$

SvS_A2
$^{87}$DDARCD$^{92}$

Cyclase domain of PaFS
$^{92}$DDVTD$^{96}$

CyS
$^{89}$DDVHCD$^{94}$

**b   1,14-Cyclization of GGPP**

(*R*)-(−)-Cembrene A (14)

ErTC-2
$^{93}$DDAIE$^{97}$

Taxa-4(5),11(12)-diene (6-8-6)

TXS
$^{613}$DDMAD$^{617}$

**c   This Work: 1,10-Cyclization of GGPP**

GGPP   n = 3
FPP    n = 2
GPP    n = 1

VenA
$^{115}$DSFVSD$^{120}$

Venezuelaene A (**1**)

Hedycaryol (**2**)

Germacrene A (**3**)

$\beta$-Farnesene (**4**)

Germacrene D (**5**)

Geraniol (**6**)

**Fig. 1 | Classification of representative type I diterpene synthases based on the initial cyclization modes.** The resolved protein crystal structures and product chemical structures of type I DTSs catalyzing the initial 1,11-cyclization (**a**) and 1,14-cyclization (**b**) of GGPP. **c** VenA-catalyzed biosynthesis of diterpene (**1**), sesquiterpenes (**2**-**5**), and monoterpene (**6**), in which the initial 1,10-cyclization of GGPP and FPP is adopted. The amino acid sequences and corresponding structures of Asp-rich motifs in the crystal structures are highlighted in red.

mechanisms for the atypical ¹¹⁵DSFVSD¹²⁰ motif and unique cyclization modes of VenA remained unsolved.

In this study, we resolved the crystal structures of VenA in both *apo* form and *holo* form in complex with a trinuclear magnesium cluster and PP$_i$ group. With the resolved structures, substrate docking analysis, molecular dynamics (MD) simulations, and hybrid quantum mechanical/molecular mechanical (QM/MM) calculations were comprehensively performed to rationalize the broad substrate spectrum and unique cyclization modes of VenA. Guided by the structure information, the amino acid residues in active site for substrate recognition/binding and carbocation stabilization and the role of ¹¹⁵DSFVSD¹²⁰ motif were explored in detail by extensive site-directed mutagenesis. Characterization of 12 terpenoids produced by different VenA mutants (but not by the wild-type enzyme) provided important mechanistic insights into the tunable substrate preference, cyclization modes, and product profiles of this bacterial DTS. Moreover, a VenA hexamutant was semi-rationally engineered as a sesterterpene synthase (STS).

## Results

### Resolution of crystal structures of VenA

To provide structural basis for the substrate promiscuity and unique cyclization mode of VenA, we first sought to crystallize VenA in its substrate-free form. However, we did not obtain any crystals of the wild-type VenA (VenA^WT) and supposed that the *N*-terminal flexible loop region (1-15 aa) might prevent the protein crystallization. Thus, the truncated form VenA^Δ1-15 with the *N*-terminal fifteen amino acids removed was expressed and purified. Of note, the in vitro enzymatic assays indicated this modification did not affect the activity of VenA (Supplementary Fig. 1, we use "VenA" to stand for "VenA^Δ1-15" unless otherwise specified). Upon this truncation, we successfully determined the high-resolution three-dimensional structure of substrate-free VenA (*apo* form) at 2.03 Å resolution (PDB ID: 7Y9H, Fig. 2a, Supplementary Table 1). In the crystal structure, VenA is dimeric, which is consistent with the result of size exclusion chromatographic analysis (Supplementary Fig. 2), indicating that VenA forms a homodimer in solution as well. The monomer of VenA holds a typical isoprenoid synthase α-fold

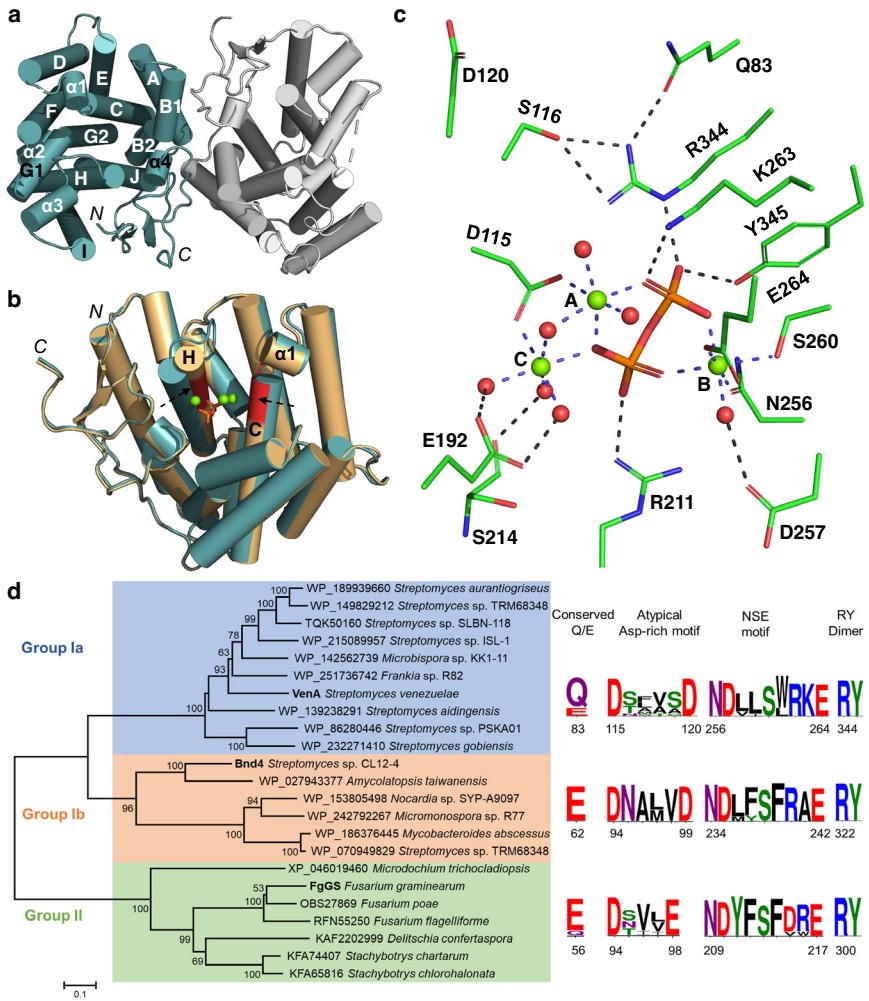

**Fig. 2 | Structures and phylogenetic analysis of VenA. a** The dimeric *apo* form of VenA. **b** The superimposition of the *apo* (teal) and PP$_i$-(Mg²⁺)$_3$-bound (brown) forms of VenA. The atypical Asp-rich motif of ¹¹⁵DSFVSD¹²⁰ on C helix and NSE motif of ²⁵⁶NDLLSWRKE²⁶⁴ on H helix are shown in red cartoon. **c** The amplified active site and key residues in the PP$_i$-(Mg²⁺)$_3$-bound VenA. The coordination and hydrogen bonds are marked by blue and black dashed lines, respectively. The water molecules and magnesium ions are shown as red and green spheres, respectively. The PP$_i$ group is shown in orange stick. **d** The phylogenetic tree of the representative microbial TPSs containing the atypical Asp-rich motif mined from NCBI databases using the neighbor-joining method with the replicates of 1000 times. Group Ia, Group Ib, and Group II are defined according to the phylogenetic relationship of the predicted TPSs with VenA, Bnd4, and FgGS, respectively. The scale bar represents 0.1 substitutions per amino acid site and the bootstrap percentages under 50% from 1000 replicates are hidden. The corresponding amino acid positions of VenA, Bnd4, and FgGS are shown below the motif of each group.

comprised of twelve core α-helices and four additional short α-helices. Helices B2, C, G1, G2, H, J, and α4 form the active site cavity, and helices A, B1, and C-terminal loop contribute to the dimer interface with a buried area of 2350.5 Å². Similar to dimeric CotB2 and SVS-A2, the two monomers are organized in an antiparallel arrangement and each active site points away from the dimer interface[29–31].

Next, we attempted to use the substrates (GPP, FPP, and GGPP) and substrate analogues (GSPP: geranyl thiopyrophosphate, FSPP: farnesyl thiopyrophosphate, and GGSPP: geranylgeranyl thiopyrophosphate) to acquire the ligand-bound structures of VenA by soaking the substrates or more stable analogues with the crystals of *apo* form. However, only the PP$_i$ group released from GGPP and a trinuclear Mg$^{2+}$ cluster were observed in the active site of VenA when soaking with GGPP, the crystal structure of which was determined at 2.18 Å (PDB ID: 7Y9G, Fig. 2b, Supplementary Table 1). Of note, when GSPP, FSPP, or GGSPP were used as substrate analogues which are supposed to resist hydrolysis; still, we only observed the hydrolyzed product thiopyrophosphate remaining in the active site. The PP$_i$-Mg$^{2+}$-bound and *apo* forms of VenA show slightly different conformations with a Cα root-mean-square deviation (RMSD) value of 0.327 Å. The obvious movements were found in C and H helices, which resulted from the binding of metal ions and PP$_i$ group by the catalytic motifs, including the atypical Asp-rich motif ($^{115}$DSFVSD$^{120}$) located on C-helix and NSE motif ($^{256}$NDLLSWRKE$^{264}$) on H-helix (Fig. 2b). These conformational changes would shield the active site from bulk solvent to avoid the premature quenching of carbocation intermediates by water[12].

## Analysis of the active site of PPi-(Mg²⁺)₃-bound VenA

In the PP$_i$-(Mg$^{2+}$)$_3$-bound form, a trinuclear magnesium cluster is coordinated in an octahedral manner by the conserved amino acid residues in the Asp-rich and NSE motifs, PP$_i$ group and water molecules (Fig. 2c). Specifically, Mg$^{2+}_A$ and Mg$^{2+}_C$ are coordinated by ten oxygen atoms of Asp115, PP$_i$ group, and surrounding water molecules on the same side of the active site. In the opposite side of the active site, Mg$^{2+}_B$ is coordinated by six oxygen atoms of NSE triad (Asn256, Ser260 and Glu264), PP$_i$ group, and one water molecule. The hydrogen bonds between coordinated water molecules and Glu192, Ser214, and Asp257 were also observed in the PP$_i$-(Mg$^{2+}$)$_3$-bound form of VenA. Upon coordination with the trinuclear magnesium cluster, Asp115 and NSE motif stretch towards the active site and lead to the movement of C and H helices, which reduces the volume of substrate binding cavity (Fig. 2b, c). Meanwhile, the PP$_i$ group forms a number of hydrogen bonds with the "PP$_i$ sensor" residues Arg211 and Lys263 in the NSE motif, and the $^{344}$RY$^{345}$ dimer in the active site of VenA, which play crucial roles in substrate trapping and conversion[12]. The "PP$_i$ sensor" Arg and RY dimer are strictly conserved among most bacterial type I TPSs for PP$_i$ binding and known as the catalytically important basic motifs[32]. Besides the amino acid residues in metal-binding motifs, Arg344 and Tyr108 in the active site of PP$_i$-(Mg$^{2+}$)$_3$-bound VenA rotate by a wide margin compared with those in *apo* form (Supplementary Fig. 3). The side chain of Tyr108 forms hydrogen bonds with the guanidine group of Arg344 in the *apo* form, but is rotated to the opposite direction in the *holo* form. This movement may reduce the spatial hindrance for the substrate entry. Meanwhile, the guanidine group of Arg344 is rotated towards the PP$_i$ group, thereby forming a hydrogen bond. Ser116 and Gln83 in the PP$_i$-(Mg$^{2+}$)$_3$-bound form may help maintaining the hydrophobic catalytic pocket by interacting with Arg344 (Fig. 2c, Supplementary Fig. 3).

To explore the key interactions observed in the active site of PP$_i$-(Mg$^{2+}$)$_3$-bound VenA, we constructed an array of mutants and analyzed their catalytic activities (Fig. 3). Unsurprisingly, the three single mutants VenA$^{N256L}$, VenA$^{S260A}$, and VenA$^{E264L}$ completely lost their activity toward GGPP, FPP, and GPP in vitro, indicating that the coordination between Mg$^{2+}_B$ and the NSE triad is required for VenA

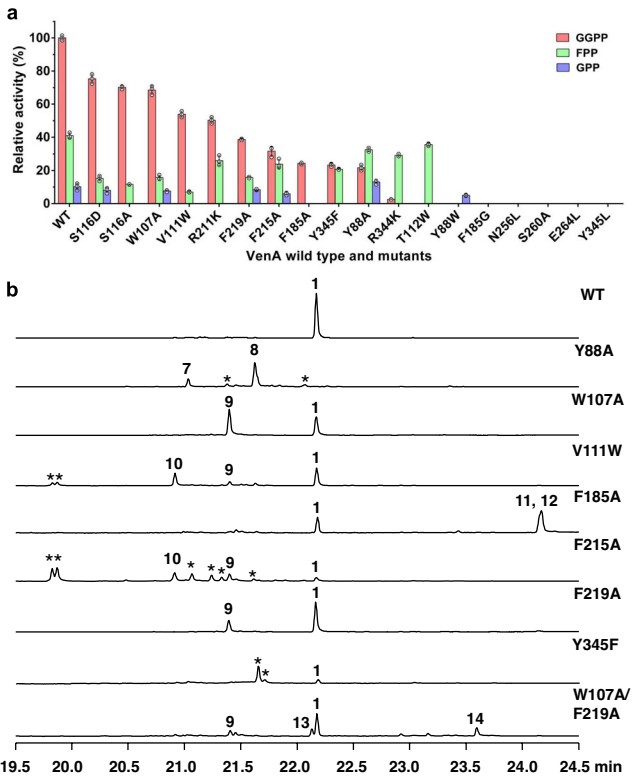

**Fig. 3 | Catalytic activities of VenA mutants. a** The relative catalytic activities of wild-type and mutant VenAs towards GGPP, FPP, and GPP. The GGPP conversion ratio of wild-type VenA is assigned as 100%. **b** GC-MS (total ion chromatograms) analysis of the diterpene products of VenA$^{WT}$ and a select number of VenA mutants. The asterisks stand for the uncharacterized diterpenes. The average values (mean ± standard deviation, $n = 3$) were used to compare the relative catalytic efficiencies between VenA wild type and mutants towards GGPP, FPP, and GPP.

catalysis. VenA$^{R221K}$ could not accept GPP and its activity toward GGPP and FPP decreased by 49.8% and 36.4%, respectively (Fig. 3a). Interestingly, the substrate preference of VenA$^{R344K}$ was shifted to favoring FPP, giving rise to two main products, **2** and germacradien-6-ol (**18**, Supplementary Figs. 4 and 5). VenA$^{Y345L}$ turned out to be a dead mutant, while VenA$^{Y345F}$ retained 23.4% and 50.8% activities toward GGPP and FPP, respectively (Fig. 3a), and produced diterpenes and sesquiterpenoids in the product ratios substantially different from that of VenA$^{WT}$ (Fig. 3b, Supplementary Figs. 4 and 5, Supplementary Table 2) perhaps due to perturbation of the hydrogen bond network. These results revealed that the hydrogen bonds established by Tyr345, Arg211, and Arg344 with the PP$_i$ group are important for substrate binding and conversion. Taken together, VenA utilizes both metal ions and basic residues to trigger the departure of the substrate PP$_i$ group, thereby generating the highly reactive allylic cation as other type I TPSs[12].

## Investigation on the atypical "Asp-rich" motif

In the past twenty-six years, a wealth of biochemical studies have demonstrated that the typical Asp-rich motif (DDXX(X)D/E) of TPSs is essential for coordinating divalent metal ions (Mg$^{2+}$/Mn$^{2+}$) to initiate substrate binding and subsequent catalysis[12,33,34]. The literature research and investigation on the Asp-rich motifs of representative crystal structures (Supplementary Fig. 6) indicate that normally two (among three) aspartates, or three all aspartates in some cases, are catalytically essential for the bacterial type I TPSs. In most bacterial type I TPSs, the first Asp in the Asp-rich motif is coordinated with Mg$^{2+}_A$ and Mg$^{2+}_C$, whereas the second Asp usually forms a salt bridge or hydrogen bond with a conserved arginine for shielding the active site

from bulk solvent, the essential role of which was characterized by site-specific mutagenesis of CotB2[29], EIZS[35], and pentalene synthase[36]. The third Asp/Glu is usually located outside of the catalytic pocket and hence dispensable for the catalytic activity, or in some cases coordinated with $Mg^{2+}_C$ as selinadiene synthase[37] (Supplementary Fig. 6).

The crystal structures of VenA are the representative structures holding an atypical DXXXXD motif, which is a counterpart of the canonical Asp-rich motif in most type I TPSs. To explore the roles of the two Asp residues, the in vitro activities of VenA[D115N] and VenA[D120N] were determined in our previous work[26]. The results clearly indicated that only the first Asp115 is required for VenA's activity; supporting this, Asp120 was found to be located outside of the active site and far from the $PP_i$ group (9.4 Å), $Mg^{2+}_A$ (7.9 Å) and $Mg^{2+}_C$ (9.1 Å) (Fig. 2c).

It is evident that the second Asp is missing from the "Asp-rich" motif of VenA. Without the second Asp in a canonical Asp-rich motif that forms a salt bridge with the conserved Arg in RY dimer of CotB2 and 1,8-cineole synthase from *Streptomyces clavuligerus*[38] (Supplementary Fig. 6), we revealed that Ser116 and Gln83 are utilized to form a hydrogen bond network with Arg344 of the conserved RY dimer in VenA (Fig. 2c, Supplementary Fig. 3). To examine the role of these interactions, the in vitro enzymatic assays of VenA[S116A] were carried out; and it entirely lost the activity toward GPP and the activities against GGPP and FPP decreased by 29.9% and 71.5%, respectively, compared with that of VenA[WT] (Fig. 3a), indicative of the importance of the hydrogen bonds between Ser116 and Arg344. The function of Ser116 might be similar to that of Asp100 in EIZS[39], which forms hydrogen bonds with the conserved Arg (Supplementary Fig. 6). Subsequently, we mutated Ser116 into Asp to restore the classical Asp-rich motif. Surprisingly, VenA[S116D] showed a decreased activity (Fig. 3a). We reason that the larger side chain of Asp might cause a steric hindrance for Arg344, thus negatively influencing the interaction of Arg344 with the $PP_i$ group. The remaining activity of VenA[S116A] might be attributed to the hydrogen bond between Gln83 and Arg344, which maintains the favorable conformation of Arg344 despite the loss of hydrogen bonds between Ser116 and Arg344 (Fig. 2c). However, we failed to confirm this hypothesis because the Gln83 mutants including VenA[Q83L], VenA[Q83K], and VenA[Q83N] formed inclusion bodies when expressed in *Escherichia coli*. These results suggest that Glu83 in B1/B2-helix-break might be critical for the correct protein folding of VenA.

Interestingly, the previous functional analysis of a similar atypical "Asp-rich" motif (94DNAMVD99) in another bacterial type I DTS Bnd4 from *Streptomyces* sp. CL12-4[40] also revealed that only Asp94 is necessary for Bnd4 activity. Based on sequence similarity network (SSN)-based homologues search, sequence alignment analysis of VenA, Bnd4 and VenA homologues collected from UniProt database, and AlphaFold2[41] predicted structures (Supplementary Figs. 7–10), we putatively proposed that Glu62 and Asn95 of Bnd4 might function as the counterparts of Gln83 and Ser116 in VenA, respectively, to form a hydrogen bond network with Arg322 of the RY dimer in Bnd4. Additionally, we observed analogous hydrogen bonds existing between Glu56, Ser95, and Arg300 of the RY dimer in the crystal structure of the fungal FgGS[42] with a corresponding motif of 94DSVLE98 (Fig. 2d, Supplementary Fig. 10). Thus, we used these three motif sequences as probes to screen the NCBI databases and revealed dozens of putative microbial type I TPSs containing the non-classical "Asp-rich" motif, which fall into two main groups (bacterial and fungal origins) upon phylogenetic analysis. Strikingly, a conserved Glu or Gln residue is unanimously located at ~30 amino acid upstream of the "DXXX(X)D/E" motif lacking the second Asp in all these proteins (Fig. 2d). These results suggest a previously unidentified subclass of type I microbial TPSs containing a "D(S/T/N)XX(X)D/E" motif together with a far but conserved Q/E residue.

## Understanding the substrate flexibility and initial cyclization modes of VenA

This study provides the important referential crystal structures of a DTS mediating the initial 1,10-cyclization of GGPP. Also, the molecular mechanisms for catalytically promiscuous type I DTSs to accept GPP, FPP, and GGPP were underexplored[32]. Thus, to understand the reaction mechanism for the specific 1,10-cyclization of GGPP as well as molecular basis for the broad substrate spectrum, we first docked GGPP/FPP/GPP into the hydrophobic reaction chamber of VenA (PDB ID: 7Y9G) using Autodock vina[43], then equilibrated enzyme-substrate complex conformations by classical MD simulations using Amber20 package[44], and finally validated the binding poses and probed the initial reaction steps by combined QM/MM calculations using the modified QChem/Tinker interface[45].

In the optimal GGPP binding pose captured by QM/MM modelling (Fig. 4a), the acyclic substrate is folded into a specific bending conformation via π-stacking interactions owing to the enriched aromatic amino acids around substrate in the active site of VenA. With regard to the reactive geranylgeranyl cation (intermediate **A**, Figs. 4b and 5), the distances from C1 to C10=C11 double bond (3.4 and 4.0 Å for C10 and C11, respectively) are shorter than those from C1 to C6=C7 double bond (4.3 and 4.1 Å, respectively) and from C1 to C14=C15 double bond (4.8 and 4.3 Å, respectively), as detected by QM/MM optimization calculations. Evidently, the initial C1 carbocation is more inclined to attack C10 than C11 due to the shorter distance to C1 (3.4 versus 4.0 Å). Besides, the orientations of *p*-orbital between C1 and C10 show a high potency for a head-to-head overlap to form σ bond (Supplementary Fig. 11). These observations are well consistent with the fact that VenA prefers to catalyzing the Markovnikov 1,10-cyclization of GGPP (Fig. 5). This is further confirmed by reaction thermodynamic and kinetic estimations based on QM/MM calculations, as summarized in Table 1. The $PP_i$ cleavage procedure was predicted to be the rate-limiting step with a barrier of 25.4 kcal/mol and endothermicity of 13.8 kcal/mol (Table 1, Supplementary Fig. 12), which is well known for type I TPSs[11,46] and the reaction energetics is similar with our previous studies for other TPSs[6,47–49]. Moreover, the subsequent 1,10-cyclization of intermediate **A** is easily attainable with zero barrier and exothermicity of −4.0 kcal/mol, while other cyclization modes (including 1,6-/1,7-/1,14-/1,15-cyclization) are impracticable owing to much higher barriers (Table 1, Supplementary Figs. 13 and 14). To further probe the preference of 1,10-cyclization reaction step, several pico-second QM/MM MD simulations were performed based on intermediate **A**, and it was found that C1 and C10 would quickly (only take about 0.3 ps) approach and stably maintain at a distance of approximately 1.8 Å (Supplementary Fig. 15).

Furthermore, the computational protocol for GGPP was also applied for FPP and GPP binding VenA models (Supplementary Fig. 16). The QM/MM predicted that the reaction barriers of $PP_i$ cleavage are 24.9 and 24.8 kcal/mol for FPP and GPP, respectively (Table 1, Supplementary Figs. 17 and 18). Similarly, both 1,6- and 1,7-cyclization modes are infeasible owing to the high reaction barriers for FPP and GPP (Table 1, Supplementary Figs. 19 and 20). That is, VenA can only catalyze GPP to release the $PP_i$ group, forming an active carbocation that would be easily quenched by a water molecule instead of further cyclization, thus leading to the final product **6** (Fig. 1c) as experimental validation. Differently, the initial C1 carbocation of farnesyl cation is more inclined to attack C10 than C11 due to the shorter distance to C1 (3.5 versus 3.8 Å), and the 1,10-cyclization is feasible with a low barrier (only 2.5 kcal/mol) for FPP (Table 1, Fig. 4c, Supplementary Figs. 17 and 21). Thus, VenA shows the cyclase activity towards FPP to generate the cyclic products **2**, **3**, and **5**, but the cyclization reaction could also be prematurely halted to yield the acyclic product **4** (Fig. 1c). Taken together, all these computational modeling results well explained the substrate specificity of VenA, which is able to accommodate GGPP/FPP/GPP to promote the initial carbocation ionization by cleaving the

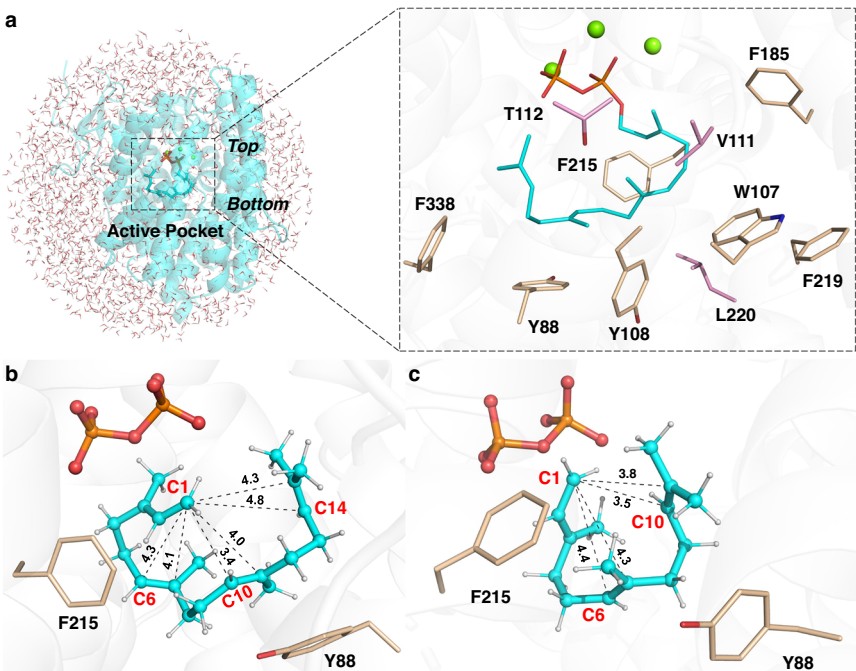

**Fig. 4 | Key structures captured by QM/MM simulations. a** The QM/MM model and zoom-in active site of VenA with GGPP binding. **b** QM/MM predicted pose of intermediate **A** as shown in Fig. 5. **c** QM/MM predicted pose of farnesyl cation. The corresponding FPP/GPP binding pose is shown in Supplementary Fig. 16. The

hydrocarbon chain and PP$_i$ of GGPP are colored in cyan and orange, respectively. The aromatic and non-aromatic amino acids are shown in wheat and pink, respectively. The water molecules are hidden to better present the zoom-in pocket (see Supplementary Fig. 16 for details).

native charged PP$_i$ group. With the solved crystal structures, these QM/MM models also provide crucial initial substrate conformations for understanding the mono-, sesqui-, and diterpene biosynthetic mechanisms of VenA.

### Elucidating the catalytic mechanism of VenA by QM/MM calculations and mutagenesis analysis of key active site residues

In the active site of TPSs, certain aromatic amino acids have been proven to be essential for stabilizing the progressively formed carbocation species, thereby directing the first carbocation intermediate to the final product via a series of C$^+$–π interactions[11,12,50]. Based on the optimal substrate binding poses of VenA as detected in QM/MM calculations and discussed above, it is promising to generate unnatural terpenoids by reprograming the aromatic residues in VenA active site. Meanwhile, the resulting terpenoid products could provide critical mechanistic insights into the cyclization process controlled by TPS. Thus, we scanned the amino acid residues within 5 Å of the GGPP in VenA, and revealed seven aromatic amino acids in addition to Tyr345 of RY dyad, including Tyr88, Trp107, Tyr108, Phe185, Phe215, Trp219, and Phe338 (Fig. 4a).

After the initial 1,10-cyclization (GGPP→intermediate **A**→intermediate **B**), to reveal the key residues involved in the following cascade reaction steps from intermediate **B** to product **1** under VenA catalysis, we conducted the reaction coordinate driven QM/MM scan calculations (Fig. 5, Supplementary Figs. 22–24). As shown in Fig. 5, the 1,3-hydride (H$_a$) transfer would overcome a reaction barrier of 12.4 kcal/mol and show a notable exothermicity of 8.9 kcal/mol. This feasible hydride transfer reaction generates a metastable intermediate, denoted as **C**. Despite being a secondary carbocation, it can be effectively stabilized by the conjugative effect with the C2=C3 double bond as well as cation-π interactions with Tyr88 and Phe215, which contribute the binding energies with the intermediate **C** of 9.6 and 6.8 kcal/mol, respectively (Supplementary Table 3). Since the distance between C1 and C14 is only 4.1 Å and their orbital orientations are well prepared, the 1,14-cyclization to append a six-membered ring in intermediate **D** is

also kinetically (9.7 kcal/mol for barrier) and thermodynamically (3.2 kcal/mol for heat release) very feasible.

Different from our previously proposed mechanism (Supplementary Fig. 25)[26], the following cyclization reaction at C15 site of intermediate **D** is triggered by C2 instead of C3. Even though the C3-C15 distance could be set as the reaction coordinate for QM/MM scan calculation, C15 is favorable for approaching C2 to construct a four-membered ring in intermediate **E**, with a barrier of 4.3 kcal/mol and heat release of 7.7 kcal/mol. Then C15 would attack C3 and make C2-C15 bond breakage to expand the four-membered ring to five-membered ring; meanwhile, C2 would attack C6=C7 double bond to yield a stable non-classical carbocation (intermediate **F**), which is termed as "protonated cyclopropane" and has been extensively studied[51–55]. The key characteristic of protonated cyclopropane is delocalization of the positive charge of the carbocation onto the three-membered ring. These steps need to conquer a barrier of 16.0 kcal/mol and release heat of 7.1 kcal/mol. Considering that the C2-C6 and C2-C7 distances are 1.8 Å and 1.7 Å (longer than a normal C−C single bond), respectively, and that the atomic charges of C2/C6/C7 are 0.49/0.25/0.08 (lower than normal C$^+$ appeared in most TPS catalysis)[12], the originally high positive charge on C2 is likely delocalized to C6 and C7. Subsequently, the structure of C2-C6-C7 protonated cyclopropane is disrupted, with C2 separating from C7 and approaching C6 to form a regular C-C single bond; as a result, the positive charge is transferred to C7 to afford intermediate **G**. This reaction procedure is kinetically very facile with a barrier of 5.8 kcal/mol. Although it shows endothermy of 4.5 kcal/mol, the highly positive C7 cooperates with the PP$_i$ group to promote the final deprotonation reaction of H$_b$ on C6, which undergoes a reaction barrier of 6.4 kcal/mol and heat release of 8.7 kcal/mol to yield the ultimate product **1**.

Taken together, the calculated four-membered ring intermediate **E** and protonated cyclopropane intermediate **F** led to a substantially revised biosynthetic mechanism for **1** (Fig. 5), which is energetically and structurally plausible and avoids the unreasonable intermediates in our previous proposal[26] (Supplementary Fig. 25). In view of the

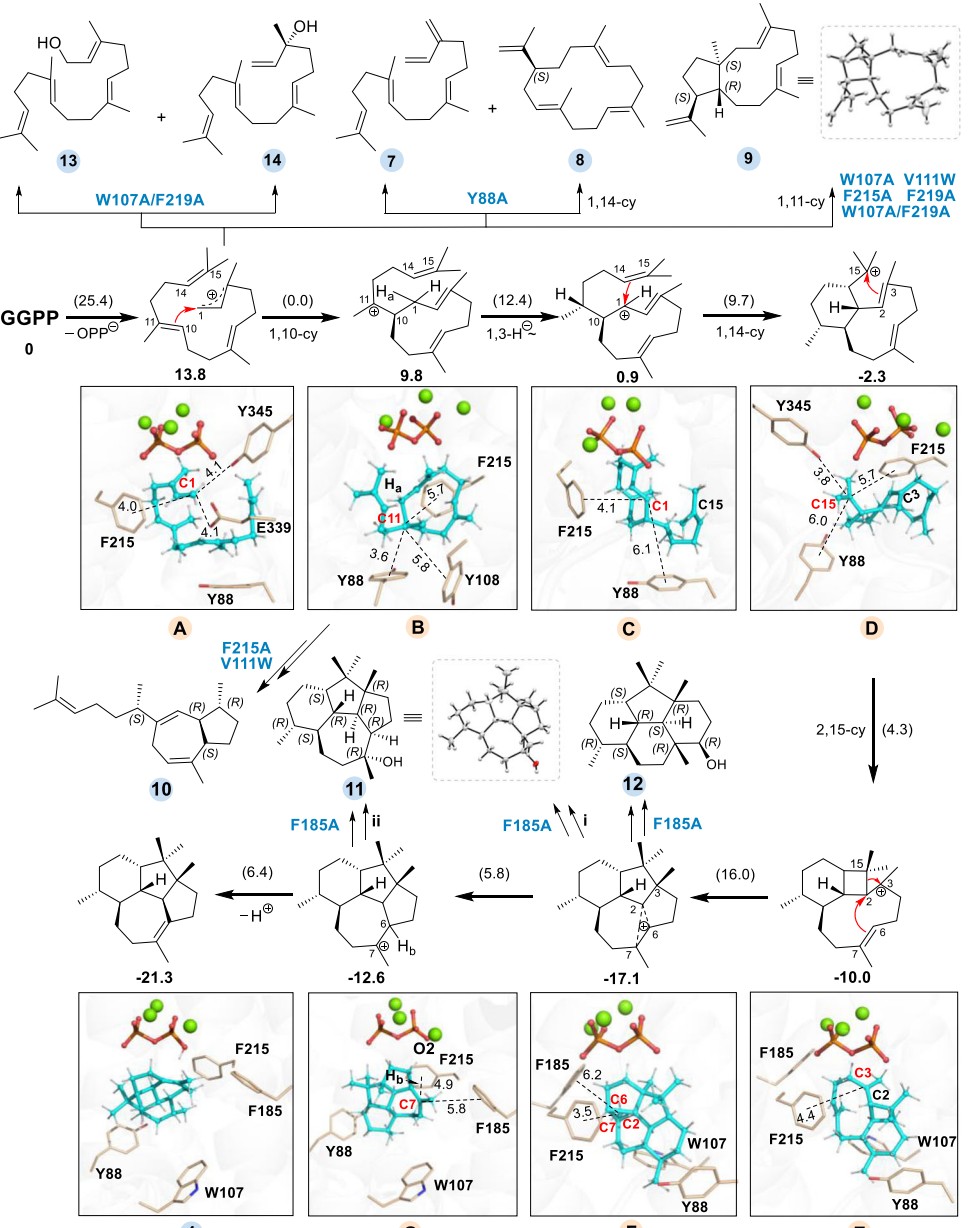

**Fig. 5 | QM/MM calculation-based reaction mechanisms for VenA^{WT} and the proposed mechanisms for other products generated by indicated VenA mutants.** The solved crystal structures of **9** and **11** are shown in dashed line frames and the QM/MM predicted structures are given in solid line frames. The putative key interactions between carbon cation intermediates and VenA residues are shown by dashed lines, and the key distances (<6.5 Å) are marked. The relative energy of each intermediate relative to substrate GGPP (in bold) and the reaction barrier for each step (in parentheses) are also given (unit: kcal/mol). Two possible biosynthetic routes (i and ii) to **11** mediated by VenA^{F185A} are shown. The water molecules are hidden to better present the zoom-in pockets.

**Table 1 | Summary of energetic values for the selected reaction steps predicted by QM/MM calculations**

|  | PP$_i$ cleavage | 1,6 cyclization | 1,7 cyclization | 1,10 cyclization | 1,14 cyclization | 1,15 cyclization |
|---|---|---|---|---|---|---|
| GGPP | 25.4/13.8 | >40 | >40 | No barrier/−4.0 | 23.5/20.4 | 22.7/18.9 |
| FPP | 24.9/23.5 | >40 | >30 | 2.5/−1.3 | - | - |
| GPP | 24.8/20.8 | >35 | >25 | - | - | - |

All values (in kcal/mol) are the reaction barriers except those after the "/" which present the endothermicity (positive values) or exothermicity (negative values). The QM/MM modeling was carried out at the M06-2X/6-31 G(d)//AMBER99SB theoretical level. More detailed computational protocols and the reaction potential energy profiles are provided in Methods and Supplementary Information.

whole catalytic reaction process, the rate-limited step is the initial PP$_i$ cleavage step with the highest barrier of 25.4 kcal/mol, and the overall heat release is about 21.3 kcal/mol (Supplementary Fig. 23). Tyr88 is mostly responsible for the first half of the reaction, while Phe185 and Trp107 are essential residues for regulating the second half of the reaction. Differently, Phe215 serves as an important residue during the whole reaction (Fig. 5, Supplementary Table 3). Of note, we took into consideration of the conformational changes of the active-site residues before and after the reaction in comparison to the crystal structure, and the low RMSD values validated the calculations (Supplementary Fig. 26).

To further investigate the function of each GGPP-surrounding aromatic amino acid (Fig. 4a) during the VenA-mediated cyclization, we performed alanine scanning for these residues. Except VenA$^{Y108A}$ and VenA$^{F338A}$ which formed inclusion bodies, other mutants were successfully expressed in *E. coli* BL21(DE3), suggesting that Tyr108 and Phe338 might be crucial for correct protein folding of VenA. According to GC-MS and NMR analyses of the purified diterpene products from the recombinant *E. coli* strain Eco-A$^{Y88A}$D co-expressing VenA$^{Y88A}$ and VenD (for overproduction of GGPP, Supplementary Table 4), the Y88A mutation led to accumulation of the acyclic product *β*-springene (**7**) and the 1,14-cyclization product (*S*)-(+)-cembrene A (**8**) (Fig. 3b, Supplementary Figs. 27–35, Supplementary Tables 2 and 5), but entirely lost the ability to produce **1**. The activity of VenA$^{Y88A}$ towards GGPP decreased by 78.3% compared with that of VenA$^{WT}$ (Fig. 3a). This result suggests that Tyr88 in B2-helix might be crucial for maintaining the conformation of intermediate **A** by steric restriction and stabilizing intermediate **B** via cation-π interactions (Supplementary Table 3), thus leading the biosynthetic route to the 1,10-cyclization product **1** (Fig. 5). Upon replacement of the bulky Tyr88 by Ala, the initial intermediate **A** with a changed conformation due to the expanded active site might undergo either direct deprotonation to give **7** or the electrophilic attacking of C14=C15 double bond by C1 cation to yield **8**. Similarly, VenA$^{Y88A}$ accepted FPP to form the acyclic product *β*-farnesene (**4**) as the main product, and small amounts of **3**, *α*-farnesene (**15**), nerolidol (**16**), and farnesol (**17**). VenA$^{Y88A}$ was unable to generate **2**, suggesting that stabilization of 11-germacryl cation requires the presence of Tyr88 (Supplementary Table 2, Supplementary Figs. 4, 5 and 16).

VenA$^{W107A}$ converted GGPP to **1** and **9** in lower efficiency compared with VenA$^{WT}$ (Figs. 3 and 5, Supplementary Table 2). Compound **9** was isolated from Eco-A$^{W107A}$D and identified as a diterpene (1*S*,3*E*,7*E*,11*R*,12*S*)-3,7,18-dolabellatriene, the titer of which reached 275.5 ± 9.4 mg/L upon 4-day fermentation in shaking flasks (Supplementary Table 6, Supplementary Figs. 36–44). Based on $^{13}$C-tracer NMR experiments, **9** was confirmed to be initiated by 1,11-cyclization of intermediate **A** (Fig. 5, Supplementary Fig. 45). With regard to the sesquiterpenoid products from FPP, the percentages of acyclic products (**4**, **16**, and **17**) of VenA$^{W107A}$ were enhanced compared with those of VenA$^{WT}$ (Supplementary Table 2, Supplementary Figs. 4 and 5). Mechanistically, VenA$^{W107A}$ prefers anti-Markovnikov 1,11-cyclization might result from the subtle changes of the GGPP's conformation in the modified active site, which might also explain the result that VenA$^{F219A}$ converted GGPP into **9** in a lower yield than the co-product **1** (Fig. 3b, Supplementary Table 2). Compared with VenA$^{WT}$, the increased total product percentage of the acyclic sesquiterpenoids (**4**, **16**, and **17**) generated by VenA$^{F219A}$ also suggests the important role of Phe219 for the 1,10-cyclization of FPP. When we combined the mutations of W107A and F219A aiming to enhance the titer of **9**, which could be used as a potential semi-synthesis precursor for bioactive agents (Supplementary Fig. 46)[56], the expected synergetic effect did not show since the double mutant VenA$^{W107A/F219A}$ produced a lower amount of **9** than each single mutant (Fig. 3b, Supplementary Table 2). Starting from either FPP or GGPP, VenA$^{W107A/F219A}$ generated the acyclic and hydrated sesqui- and diterpenoids, including geranyllinalool (**13**), geranylgeraniol (**14**), **16** and **17**, likely owing to the significantly enlarged and hence leaking active site (Figs. 3b and 5, Supplementary Figs. 4 and 5, Supplementary Table 2).

The mutation of Phe185 in F-helix into Ala led to two monohydroxy diterpenoids together with 25.8% of **1**, including **11** (named as venezuelaenol A) with the same 5-5-6-7 skeleton of **1**, and the main product **12** (named as venezuelaenol B, *see* Supporting Information for details) with a unique 6-5-6-6 skeleton, in which the five-membered ring is merged with all three six-membered rings (Figs. 3 and 5, Supplementary Figs. 47–63, Supplementary Tables 2 and 7-8). Of note, VenA$^{F185A}$ appeared as a strict DTS (24.4% activity of VenA$^{WT}$) without any detectable activity towards GPP and FPP (Fig. 3a). When Phe185 was mutated into the smallest amino acid Gly, the mutant VenA$^{F185G}$ was completely inactivated (Fig. 3a). These results strongly suggest that Phe185 is an essential gatekeeper to keep strict hydrophobic environment in the closed form of VenA. Once this barrier is broken by changing it into a smaller amino acid residue, the entrance of water molecule(s) would disrupt the hydrophobic environment and hydrogen bond network to influence substrate binding and to prematurely quench the carbocation intermediates. Evidently, **11** could be formed from the quenching of intermediate **G** by water. Similarly, **12** is proposed to be generated from quenching of intermediate **F** at C6 site by water. Considering the delocalized nature of the non-classical carbocation intermediate **F**, we hypothesized that **11** could be alternatively derived from the quenching of protonated cyclopropane at C7 site by water. (Fig. 5, Supplementary Fig. 64). Taken together, Phe185 is important for stabilizing intermediates **F** and **G** during the biosynthesis of **1** together with Phe215 (Supplementary Table 3).

As previously reported[30,37], the amino acid residues located in the G1/2 helix-break motif of type I bacterial TPSs are usually non-aromatic due to the flexibility in this region (Supplementary Fig. 9), which are considered catalytically important for selinadiene synthase from *Streptomyces pristinaespiralis*[37]. By contrast, an aromatic amino acid residue Phe215 is located in the G1/2 helix-break region of VenA (Fig. 4, Supplementary Fig. 9). When Phe215 was mutated into Ala, the diterpene product diversity was significantly improved compared with that of VenA$^{WT}$ according to the in vitro enzymatic assays (Fig. 3b). However, we did not structurally identify all the diterpene products due to the cost-prohibitive isolation of these compounds in very low yields. The two main products of VenA$^{F215A}$ were determined to be **9** and **10** (named as dictytriene C, Figs. 3 and 5, Supplementary Figs. 65–73, Supplementary Table 9). Of note, **10** is the representative diterpene with a 5-7 skeleton assembled by a bacterial DTS, which was confirmed to be initiated by 1,10-cyclization of intermediate **A** based on our $^{13}$C-NMR tracer experiments (Fig. 5, Supplementary Fig. 74). These results combined with the above-described MD and QM/MM simulation results indicate that Phe215 should be involved in the stabilization of all the intermediates **A**–**G** (Fig. 5, Supplementary Table 3). When using FPP as substrate, VenA$^{F215A}$ synthesized the acyclic **4**, **16**, and **17** as main products, suggesting that Phe215 might participate in stabilizing 11-germacryl cation; this is also supported by our QM/MM simulations (Supplementary Table 2, Supplementary Figs. 4, 5 and 16).

## Structure-guided engineering of substrate specificity

A few type I DTSs have been identified to show considerable substrate promiscuity for polyprenyl phosphate substrates with different chain-length, and even totally unexpected prenyltransferase activities for small molecules[42,57,58]. Previously, the substrate specificities of such DTSs (e.g., SvS-A2) could be changed through screening the active-site residues by site-directed mutagenesis[30,59]. However, due to paucity of structural information for the substrate-, substrate analogue-, or Mg$^{2+}$/Mn$^{2+}$-PP$_i$-bound forms of bacterial type I DTSs, it is still challenging to rationally engineer the substrate specificity of these DTSs. Herein, VenA with significant substrate promiscuity and resolved crystal structures provides an ideal DTS to practice the structure-guided engineering of substrate specificity.

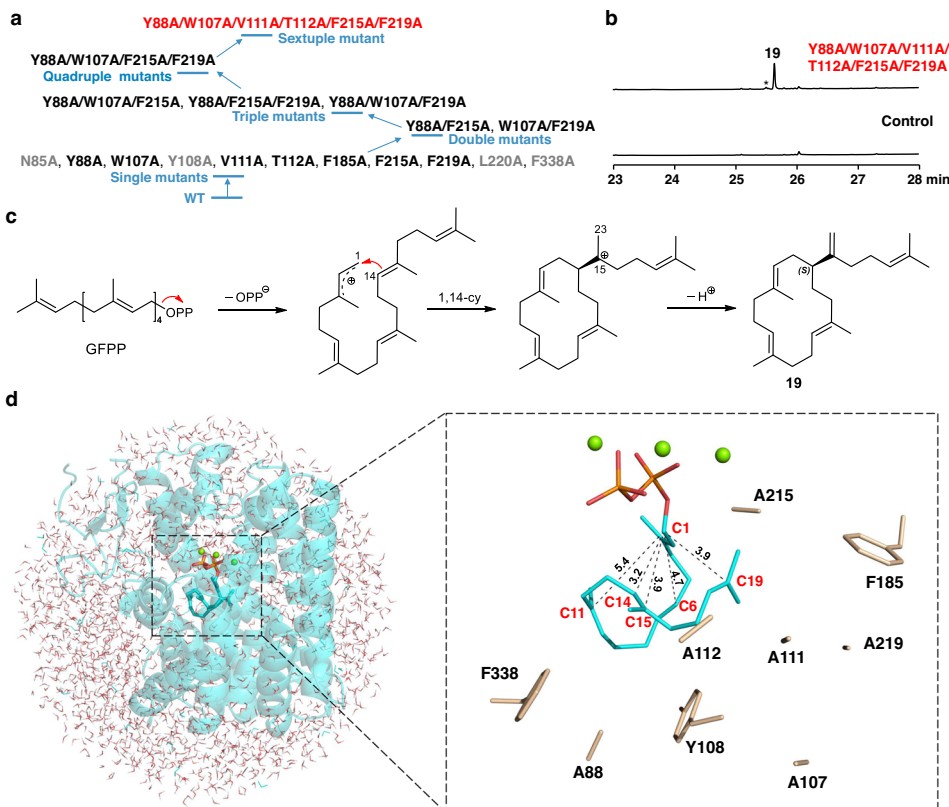

**Fig. 6 | Development and analysis of the sesterterpene synthase VenA$^{Y88A/W107A/}$** **$^{V111A/T112A/F215A/F219A}$. a** The summary of the protein expression state and sesterterpene biosynthetic activity of VenA mutants. The mutants marked by grey, black, and red colors stand for the insoluble mutants, soluble but inactive mutants, and soluble and active mutants, respectively. **b** GC-MS analysis of the in vitro activity of the sextuple VenA mutant towards GFPP. The asterisk stands for the uncharacterized sesterterpene. **c** The proposed biosynthetic mechanism of **19**. **d** The QM/MM model and zoom-in active site of VenA$^{Y88A/W107A/V111A/T112A/F215A/F219A}$ with GFPP binding. Only the shorter distances from C1 to C6=C7, C10=C11, and C18=C19 double bonds are shown. The water molecules are hidden to better present the zoom-in pocket (see Supplementary Fig. 16 for details).

We collected a number of representative structures of di-, sesqui- and monoterpene synthases from the PDB database and calculated the average volumes to accommodate the prenyl chains of GGPP (~526 Å³), FPP (~438 Å³) and GPP (~372 Å³), which suggest the possibility of engineering a DTS to favor FPP or GPP by rationally reshaping the active site[59] (Supplementary Table 10). Thus, we sought to manipulate the key residues near the top and bottom of the substrate binding pocket. To change the substrate preference of VenA towards FPP or GPP, we used the largest proteinogenic hydrophobic amino acid Trp to scan the four residues within 4 Å range of GGPP, including Tyr88, Val111, Thr112, and Leu220 (Fig. 4a, *see* the different visual angle in Supplementary Fig. 75). As expected, VenA$^{Y88W}$ became a strict monoterpene synthase, specifically catalyzing the reaction of GPP to **6** likely due to the steric hindrance and reduced volume of active pocket resulted from the larger side chain of Trp (Fig. 3a, Supplementary Table 2). VenA$^{T112W}$ became a strict sesquiterpene synthase by converting FPP to the two main products **5** and **18** (Fig. 3a, Supplementary Figs. 4 and 5). These results suggest that the steric hindrance endowed by Trp on top of the active pocket could restrict the size of substrates (Supplementary Fig. 75). The sesquiterpenoid products of VenA$^{V111W}$ were the same as those of VenA$^{WT}$, but this mutant could not accept GPP and showed decreased activities towards GGPP and FPP (Fig. 3, Supplementary Fig. 4, Supplementary Table 2). Surprisingly, the diterpene product diversity was expanded by the V111W mutation (Fig. 2b). As for VenA$^{L220W}$, we were unable to test its activity since this mutant formed an inclusion body in *E. coli*.

**Semi-rational engineering of VenA into a sesterterpene synthase**

Sesterterpenoids, mainly isolated from plants, fungi, and some marine organisms, represent a rare structural class of terpenoids[1]. The reports of bacterial sesterterpenoids and STSs are even fewer[60]. Given our understanding of the molecular basis for VenA's substrate specificity, we sought to engineer this DTS into a STS, which, to the best of our knowledge, is an unprecedented challenge. To enlarge the active pocket to accept the C$_{25}$ hydrocarbon chain of GFPP, thorough alanine scanning of the active site residues was performed. Unfortunately, all the soluble single mutants showed no activity toward GFPP (Fig. 6a), which was chemically synthesized by following the established protocols[21,61] due to its commercial unavailability (Supplementary Figs. 76–78). Next, a number of double mutants (VenA$^{Y88A/F215A}$ and VenA$^{W107A/F219A}$), triple mutants (VenA$^{Y88A/W107A/F219A}$, VenA$^{Y88A/W107A/F215A,}$ and VenA$^{Y88A/F215A/F219A}$), and quadruple mutant (VenA$^{Y88A/W107A/F215A/F219A}$) were constructed by combining the alanine mutations of the key aromatic residues that would not cause protein precipitation. Again, all these multiple mutants were inactive towards GFPP. We reasoned that these active pockets might not be large enough to accommodate GFPP. Thus, we elected to mutate all the mutation sites (except Phe185) that resulted in soluble proteins into alanine at once, thereby making a very spacious active site. Notably, Phe185 was not included due to its gatekeeper role in maintaining the hydrophobic chamber as described above. As a result, the sextuple mutant VenA$^{Y88A/W107A/V111A/T112A/F215A/F219A}$ was successfully expressed and purified from *E. coli* BL21(DE3). To our delight, this hexamutant converted GFPP into a main product **19** (Fig. 6b, accounting for 92.6% of total sesterterpene products). The

product was purified from in vitro enzymatic reactions and identified as a known sesterterpene (S)-cericerene containing a 14-membered macrocyclic ring (Fig. 6c, Supplementary Figs. 79–87, Supplementary Table 11)[62].

To understand the cyclization mechanism of the VenA-derived STS, GFPP was modelled into its active site. Not surprisingly, the active site volume of this hexamutant (641.5 Å³) is much larger than that of VenA^WT (516.1 Å³). The optimal binding conformation of GFPP well supports an initial C1−C14 cyclization since the distance from C1 to C14 (3.2 Å) is shorter than those from C1 to C15 (3.9 Å), C6=C7 (4.7 & 5.6Å), C10=C11 (6.0 & 5.4Å) and C18=C19 (3.9 and 3.9 Å) double bonds (Fig. 6d). Mechanistically, the biosynthesis of **19** could be rationalized by enzymatic abstraction of $PP_i$ group, the carbocation-initiated intramolecular C1−C14 bond formation, and direct deprotonation to form the C15−C23 double bond (Fig. 6c).

## Discussion

At present, most of identified bacterial TPSs are sesquiterpene synthases[32]. Comparatively, the functions, structures, and mechanisms of DTSs that show higher functional complexity and plasticity, and product diversity are much less studied[63,64]. A phylogenetic tree built from VenA and 26 identified bacterial type I DTSs reveals a scattered distribution of DTSs with different cyclization modes (Supplementary Fig. 88), indicating that the prediction of function and mechanism from primary amino acid sequence of a bacterial DTS is difficult. This further emphasize the significance of elucidating the structure-function relationship of bacterial DTSs for understanding the mechanisms behind the highly diversified cyclization modes. Although de novo protein folding tools have shown promising prospects for (semi)rational design of TPSs[41], precise prediction of the positions of metal ions and polyprenyl pyrophosphate substrates remains challenging. Thus, more real structures of TPSs are still required to be resolved to further our mechanistic and engineering studies. However, to the best of knowledge, there have been only three reported crystal structures of bacterial type I DTSs that use GGPP as the substrate, including CotB2[29,31], the reconstructed ancestor of SvS (SvS-A2)[30] and CyS[65]. Since the length of CotB2 (307 aa) is significantly shorter than most identified type I DTSs (340–390 aa) and the structural information for the *holo* form of SvS-A2 is unavailable, we propose that the VenA and CyS structures could serve as structure-modeling templates to guide the future enzyme engineering efforts for other bacterial DTSs.

This study provides significant structural insights into a bacterial type I TPS containing an unusual ^115DSFVSD^120 motif. Based on bioinformatics and mutagenesis analyses, as well as the analogous hydrogen bond networks (e.g., Ser116-Arg344-Gln83 in VenA) observed in the crystal structures of VenA, FgGS and the predicted structure of Bnd4 (Fig. 2c, Supplementary Fig. 10), we propose a subclass of type I microbial TPSs containing the "D(S/T/N)XX(X)D/E" motif and a conserved Glu/Gln residue -30 amino acid upstream. These findings suggest that the previously neglected Glu/Gln might have co-evolved with the atypical "Asp-rich" motif in both bacterial and fungal TPSs, thus lowering the selection pressure for the second Asp in the canonical Asp-rich motif. Aside from the variability of the second and third Asp in the atypical "Asp-rich" motif of VenA and Bnd4[40] (Supplementary Fig. 10), recently, the sesterterpene synthase SmTS1 from *Streptomyces mobaraensis* was predicted to have an unusual ^86NDLTV^90 motif, of which the first Asp is replaced by Asn[60,66]. These natural alterations not only demonstrate the evolvability of the conserved Asp-rich motif in bacterial type I TPSs, but also suggest the possible existence of some bacterial type I TPSs that may contain a single or even none Asp in an "unrecognizable" $Mg^{2+}$ binding motif. This finding will complicate, but eventually benefit the future bioinformatics-based mining of novel TPSs and terpenoids.

The initial cyclization is the most decisive step for the catalysis mediated by type I TPSs. To the best of our knowledge, only six structures of type I DTSs using GGPP as the substrate from plant, fungi, and bacteria have been elucidated prior to this study, which catalyze the initial 1,11-cyclization (CotB2, SvS-A2, CyS, and PaFS[67]) and 1,14-cyclization of GGPP (taxadiene synthase[68] and ErTC-2[69]) (Fig. 1a, b, Supplementary Table 12, Supplementary Fig. 89). On the basis of the crystal structure resolution of a DTS (i.e., VenA) that initiates 1,10-cyclization of GGPP, relevant multiscale simulations, and site-directed mutagenesis analyses, a more reasonable biosynthetic mechanism for construction of the unique 5-5-6-7 tetracyclic skeleton in **1** was proposed (Fig. 5).

Guided by structural information, the mutation of key amino acids in VenA could turn this DTS to prefer 1,11- or 1,14-cyclization of GGPP, displaying the functional plasticity of VenA. In particular, VenA^Y88A was found to favor 1,14-cyclization of GGPP. Together with the predicated key residues in the active pockets of several identified bacterial type I DTSs (Supplementary Table 13), we propose that the aromatic amino acid in B-helix might be responsible for directing the C1 carbonation to attack C10=C11 or C14=C15 double bonds. This sequence characteristic in combination with the structure(s) of diterpene product(s) may help predicting the initial cyclization mode by type I DTS, which however requires more experimental evidences. Despite recent progresses from this and other laboratories on TPSs[70,71], it is still difficult to predict the initial cyclization mode and final product structures simply based on protein sequence analysis. We envision that the growing understanding of the mechanisms and structure-function relationships of TPSs, together with the AlphaFold2[41]-based TPS structures prediction and fast-advancing machine learning technologies may eventually accomplish this "impossible mission" in the future.

By replacing each GGPP-surrounding residues with a Trp or Ala, we successfully engineered a geraniol synthase (VenA^Y88W), a sesquiterpene synthase (VenA^T112W), and a DTS with higher selectivity for GGPP (VenA^F185A) than VenA^WT. Together with structural information, these mutant TPSs with different substrate specificity provide important clues for understanding the molecular mechanisms for the substrate promiscuity of VenA. In particular, we successfully converted VenA into an STS by expanding the active pocket to a large extent, which presents an example of engineering a DTS to accept the larger unnatural substrate GFPP. These results suggest that semi-rational modification of the active site volume by using differently sized hydrophobic amino acids could be an effective strategy for engineering substrate specificity and expanding substrate scope of TPSs[62,66]. It is worth noting that the steric hindrance would sometimes lead to undesired perturbations in the active site, thus unexpectedly affecting substrate recognition and conversion.

Chemical synthesis of polycyclic terpenes in a regio- and stereoselective manner is challenging and the traditional isolation strategy from natural resources is unsustainable and suffers from low yields and high costs[72]. TPSs catalyzing the most complex cyclization reactions of polyprenyl pyrophosphates in nature under mild conditions override many chemical catalysts[6], indicating that the development of biocatalysts based on structural and mechanistic information of TPSs is promising. Thus, construction of microbial cell factories based on the rewired polyprenyl pyrophosphate biosynthetic pathways and engineered TPSs could provide an alternative cost-effective way for production of high-value-added terpenoids[73]. The VenA mutants showing different product portfolios enrich the synthetic biology toolbox and would enable new approaches for production of the diterpenes as bioactive agents or useful (bio)synthetic precursors. For instance, commercial fragrances or bioactive terpenoids (**4**, **6**, **7**, **13–17**, and **19**), the biosynthetic precursors for diverse bioactive terpenoids (**2**, **3**, **5**, **8**, and **18**), and the unusual diterpenoids (**9–12**) could be prepared by wild-type or certain mutant VenA enzymes. Among these,

**10** and **11** showed moderate inhibitory activities to a select number of Gram-positive and Gram-negative bacteria with some of which are plant pathogens (Supplementary Table 14), suggesting the application potential of 5-7 and 5-5-6-7 skeleton terpenoids in pharmaceutical and agricultural industries. It is noteworthy that hundreds of dolabellane-type natural products have been isolated from metazoans and plants, which show diverse bioactivities including antibacterial, anti-inflammatory, antifeedant, antivirus, and cytotoxic activities (Supplementary Fig. 46), thus holding great application potentials in pharmaceutical industry[56]. The highest titer (275.5 mg/L) of microbial dolabellatriene skeleton reported in this study could provide a low-cost and sustainable resource for preparation of useful derivatives by semi-synthesis or biotransformation.

During submission of this work, a remarkable paper on Bnd4 mutagenesis was published[74]. Interestingly, some Bnd4 mutants also generate **7**, **13**, and **14** as VenA mutants. VenA$^{F338A}$ is insoluble while the corresponding mutant of Bnd4$^{W316A}$ is soluble and active. VenA$^{F185A}$ generates the mono-hydroxy diterpenoids **11** with a 5-5-6-7 skeleton and **12** with a 6-5-6-6 skeleton, but the counterpart mutant Bnd4$^{F162A}$ yields acyclic diterpenoids. These results further demonstrate the big challenge of the rational design of TPSs.

In summary, we elucidated the structural basis for the substrate promiscuity and initial cyclization mechanisms of the bacterial type I DTS VenA, which will benefit the prediction and understanding of the catalytic functions of VenA homologues and other DTSs from bacteria (Supplementary Fig. 7). In the crystal structure of the PP$_i$-(Mg$^{2+}$)$_3$-bound form of VenA, we revealed a number of crucial hydrogen bond networks (PP$_i$–Arg211/Lys263/Arg344/Tyr345–H$_2$O) and coordination bonds (Mg$^{2+}_A$–Asp115–Mg$^{2+}_C$–PP$_i$–H$_2$O and Mg$^{2+}_B$–Asn256/Ser260/Glu264–PP$_i$–H$_2$O). The role and mode of action of the atypical "Asp-rich" motif ($^{15}$DSFVSD$^{120}$, unlike the canonical DDXX(X)D/E motif) in VenA was determined by structural analysis and structure-guided mutational analysis in detail. The mechanisms for the broad substrate spectrum, initial 1,10-cyclization mode, and following complex cyclization process of VenA were elucidated by molecular docking, MD, and QM/MM analyses. We also explored the functions of the key residues in VenA for carbocation stabilization and maintenance of the hydrophobic contour of the active site. As an additional result, four unusual diterpenoids were obtained during our tuning of the cyclization modes of VenA. Furthermore, manipulation of a number of specific residues reshaped the substrate preference and released the catalytic potential of VenA by tuning the volume and shape of the active pocket and hence the substrate binding conformation.

# Methods
## Materials
All chemicals and antibiotics used in this study were obtained from Cayman, Sigma Aldrich, or Sinopharm Chemical Reagent, unless otherwise specified. Fast-digest restriction endonucleases were purchased from Monad. Isolation of plasmid DNA from *E. coli* was performed using the MonPure™ Plasmid Mini Prep Kit from Qingdao Baisai Biotechnology. Purification of DNA fragments from PCR reactions or agarose gels was performed using a MonPure™ Gel & PCR Clean Kit from Qingdao Baisai Biotechnology. ClonExpress Ultra One Step Cloning Kit was purchased from Vazyme. Spark HiFi Seamless Cloning Kit was bought from Shandong Sparkjade Biotechnology Co., Ltd. I-5™ 2× High-Fidelity Master Mix and 2 × T5 Super PCR Mix (Colony) was purchased from Tsingke Biological Technology. *n*-Alkanes (C10-C40) standard mixture was bought from O2Si. Ni-NTA Sefinose™ Resin (Settled Resin) for protein purification was purchased from Sangon Biotech. The FlexiRun™ premixed gel solution for SDS-PAGE, GelRed, Luria-Bertani broth (LB), and Terrific broth (TB) were obtained from MDBio. Primer synthesis and DNA sequencing were performed by Tsingke Biological Technology.

## Protein crystallization and diffraction data collection
Initial crystallization screening of VenA$^{WT}$ and VenA$^{\Delta1-15}$ was performed with >1000 different reservoir compositions from Hampton research (Laguna Niguel, California, USA), Rigaku Corporation (Akishima-shi, Tokyo), and Molecular Dimensions (Newmarket, England, United Kingdom). All the crystallization experiments were conducted at 25 °C, using the sitting-drop vapor-diffusion method. In general, 1 μL protein solution (at a concentration of 20 mg/mL) was mixed with 1 μL reservoir solution in 96-well Cryschem plates (Hampton Research) and equilibrated against 80 μL reservoir solution. Initial crystals of VenA$^{\Delta1-15}$ were obtained within 7 days using the BCS Screen 1 Tubes 16 (0.1 M HEPES [pH = 7.5] and 25% PEG Smear Medium). The cryo-protectant contained 0.1 M HEPES [pH = 7.5], 25% PEG Smear Medium, 10% ethylene glycol, and 20 mM MgCl$_2$. Soaking apo form crystals in cryo-protectant containing 5 mM substrate/analogues (geranyl diphosphate, GPP; geranyl S-thiopyrophosphate, GSPP; farnesyl pyrophosphate, FPP; farnesyl S-thiopyrophosphate, FSPP; geranylgeranyl diphosphate, GGPP; or geranylgeranyl S-thiopyrophosphate, GGSPP), and venezuelaene A (**1**) was performed for at least 30 min. Moreover, prior to data collection, crystals were mounted in a cryo-loop and flash cooled with liquid nitrogen. All data sets were collected using an in-house X-ray diffractometer (Bruker D8 VENTURE of SX-XRD) of Hubei University. Data were processed by using the program PROTEUM3.

## Structure determination and refinement of VenA
The crystal structure of VenA was solved by molecular replacement with the Phaser program[75] of CCP4i using terpene cyclase SvS-A2[30] from *Streptomyces* sp. CWA1 (PDB ID: 6TBD, 23% identity) as a search model. The initial model was further optimized by using AutoBuild in PHENIX[76]. Prior to structure refinement, 5% of randomly selected reflections were set aside for calculating $R_{free}$ as a monitor of model quality. Model adjustment and refinement were carried out using Refmac5[77] and Coot[78]. All graphics for the protein structures were prepared using the PyMOL program (http://pymol.sourceforge.net/). Data collection and refinement statistics are summarized in Supplementary Table 1.

## Molecular docking
Auotodock vina software[43] was used for the molecular docking analysis of VenA with GPP, FPP, GGPP, and GFPP. Reagents and water molecules in PP$_i$-(Mg$^{2+}$)$_3$-bound VenA wild type (PDB ID: 7Y9G) and VenA$^{Y88A/W107A/V111A/T112A/F215A/F219A}$ were removed in Pymol before docking except for the water molecules directly coordinated with the trinuclear magnesium cluster. The center grid box was set as center_$x$ = 37.757, center_$y$ = −38.968, and center_$z$ = 4.867. The rest parameters remained as their default values by following the official protocol[43]. Considering several chiral centers in products, we manually checked these docking poses to avoid any irrational conformations. The optimal pose for each substrate was used as the initial structure for the following classical MD simulations.

## Classical MD simulations
The Amber FF99SB[79] force field was employed for the protein and the TIP3P[80] model was used for solvent water molecules. Meanwhile, the restrained electrostatic potential (RESP)[81] charges of ligands were calculated at the HF/6-31(G)* level using the Gaussian 16 package[82]. The force field parameters of ligands were generated from the AMBER GAFF force field[83]. The MD simulations were carried out using the Amber20 package[44], and the periodic boundary condition with cubic models were employed to all systems. The initial coordinates and topology files were generated by the tleap modules in Amber20. Sodium ions were utilized to neutralize these systems. Three steps of minimization were taken to relax the solvent molecules and protein-ligand complexes. First, only water molecules in the systems

were minimized, then side chains of residues, and finally all atoms were minimized. After minimization, each system was heated from 0 K to 300 K gradually under the NVT ensemble, followed by another 100 ps NPT ensemble MD simulations for equilibration at a constant temperature of 300 K and a pressure of 1.0 atm. Afterwards, a 30 ns NVT ensemble production MD simulation with a time step of 2 fs was performed for each model. During these simulations, general restraints were employed to keep the conformation of these substrates owing to the well-known poor treatment of metal-ligand coordination interaction and ligand conformation by force field. The SHAKE[84] algorithm was used for constraint hydrogens during MD simulations. A cutoff of 12 Å was set for both van der Waals and electrostatic interactions. After these systems were stable, snapshots of each system were chosen to build the initial structures for the subsequent QM/MM simulations.

## QM/MM simulations

All QM/MM models were prepared by deleting the solvent beyond 30 Å from the C6 atom of these substrates based on above protein-substrate complexes equilibrated by classical MD simulations. As for the selection of the QM region, all substrate atoms, the three $Mg^{2+}$ atoms together with aromatic residues Phe215 and Tyr88 were considered as the QM region during the potential energy scan (PES) processes from GGPP to intermediate **D**, since these two residues are close to the substrates and likely to stabilize the carbon cations. The Phe185 and Trp107 residues were appended into the QM region during the subsequent PES processes because the carbocations were approaching the two residues as the reaction progressed. During the QM/MM MD process based on intermediate **A** (Fig. 5) which was captured from the first PES process on GGPP, the $PP_i$ with its coordinated $Mg^{2+}$ atoms were removed from the defined QM region since this strongly charged group would not be involved in the subsequent cyclization reaction after $PP_i$ cleavage in our simulation process. Only GFPP and three coordinated $Mg^{2+}$ atoms were included in the QM region for optimizing the GFPP model, since the critical aromatic residues had already been substituted to alanine. All atoms in the QM subsystem were treated by M06-2X[85,86]/6-31 G(d) method, which is widely used in studies of cyclization reactions[87,88]. The QM/MM boundary was treated by the improved pseudo approach[89–91]. The Amber FF99SB force field was used for all the remaining atoms as in previous classical MD simulations. Spherical boundary condition was employed, and the atoms more than 30 Å from the spherical center were fixed. A 12 Å cutoff was employed for van der Waals and 18 Å for electrostatic interactions. The system temperature was controlled by Langevin thermostat method[92] at 300 K, and the Newton equation of motion was integrated by the Beeman algorithm[93] during the QM/MM MD process. The prepared structures were minimized by the QM/MM method. These optimized structures were used to map out the PES curves with the reaction coordinate driving method[94,95]. All these QM/MM calculations were performed with the modified QChem[96] and Tinker[97] programs developed by Prof. Yingkai Zhang's group (http://www.nyu.edu/projects/yzhang/). And the similar QM/MM protocol had also been employed in our previous studies of other TPSs[6,47,98].

## In vitro enzymatic assay

To test the in vitro activity of VenA wild type or mutants, a standard 50 µL reaction mixture containing 2 µM enzyme, 100 µM substrate (GPP, FPP, GGPP or GFPP), 10 mM $MgCl_2$ in Tris-HCl buffer (50 mM, 10% glycerol, pH = 7.4) was incubated at 30 °C for 2 h. The boiling-inactivated enzyme was added to the control experiment. The in vitro enzymatic reaction was quenched by adding a two-fold volume of ethyl acetate, vortexed for 10 min, and centrifuged at 14,000 × g for 10 min. Subsequently, the organic extract was taken and directly used for GC or GC-MS analysis.

## $^{13}$C-tracer NMR experiments

To prepare the randomly $^{13}$C-labelled **9** and **10** at eight odd-numbered carbons (C1, C3, C5, C7, C9, C11, C13, and C15) in vivo (Supplementary Fig. 90), a 0.5 L TB culture of Eco-A$^{VIIIW}$D and Eco-A$^{W107A}$D were individually fed with 0.4 g/L (1-$^{13}$C)-$CH_3COONa$ at the time point of protein expression induction. Similarly, to prepare the $^{13}$C-labelled **9** and **10** at the left twelve carbons (C2, C4, C6, C8, C10, C12, C14, C16, C17, C18, C19 and C20), a 0.5 mL TB culture of Eco-A$^{VIIIW}$D and Eco-A$^{W107A}$D were individually fed with 0.4 g/L (2-$^{13}$C)-$CH_3COONa$ when inducing the expression of proteins. After 4 d fermentation at 18 °C, the fermentation broth was extracted by an equal volume of ethyl acetate three times, and the $^{13}$C-labelled **9** and **10** (the mixture of isotopomers) were purified for $^{13}$C-NMR analysis (Supplementary Figs. 45 and 74).

## Reporting summary

Further information on research design is available in the Nature Portfolio Reporting Summary linked to this article.

## Data availability

The structural data of *apo* form and $PP_i$-$(Mg^{2+})_3$-bound form of VenA generated in this study have been deposited in the Protein Data Bank (PDB) database under accession codes 7Y9H and 7Y9G, respectively. The crystal structure of SvS-A2 used in this study is available in the PDB database under accession code 6TBD. The structural data of **9** and **11** generated in this study have been deposited in the Cambridge Crystallographic Data Centre (CCDC) under accession codes 2163519 and 2184782, respectively. All other relevant data supporting the findings of this study are available in this manuscript and Supplementary Information.

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

## Acknowledgements

This work was supported by the National Key Research and Development Program (2021YFA0911500 to L.D. and 2019YFA0905100 to S.L.), the National Natural Science Foundation of China (32200017 to Z.L., 32025001 to S.L., 32000039 to X.Z. and 32170088 to L.D.), the China National Postdoctoral Program for Innovative Talents (BX20220191 to Z.L.), the China Postdoctoral Science Foundation (2021M701989 to Z.L. and 2022M710080 to Y.J.), the Shandong Provincial Postdoctoral Innovation Project (SDCX-ZG-202201005 to Y.J.), and the Shandong Provincial

Natural Science Foundation (ZR2019ZD20 to S.L. and ZR2022QC070 to Z.L.). The authors thank Guannan Lin, Jing Zhu, Zhifeng Li, and Haiyan Sui from the State Key Laboratory of Microbial Technology, Shandong University for their assistance in the collection of HRMS, CD, GC-MS, NMR, and single-crystal X-ray diffraction data. We also thank Professor Dehai Li at the Ocean University of China for his assistance in ECD analysis.

## Author contributions

Z.L., R.-T. G., R.W., and S.L. conceived this research. Z.L. performed the majority of biochemical experiments, and the isolation and structural elucidation of terpenoids. L.Z. screened crystals and solved the crystal structure of VenA. K.X. performed the multiscale simulations. Y.J., J.D., Q.W., Y.H., X.J., and Z.X. helped in different experiments. X.Z., L.-H.M., L.D., and X.L. assisted in data collection and NMR analysis. Z.L., L.Z., K.X., Y.-J.T., R.-T. G., and S.L. analyzed experimental results. Z.L., L.Z., K.X., R.W., R.-T. G., and S.L. wrote the manuscript. All authors read and approved the manuscript.

## Competing interests

The authors declare no competing interests.
