## [Peer Review File · Nature Communications]

Molecular insights into the catalytic promiscuity of a bacterial diterpene synthaseREVIEWER COMMENTS

Reviewer #1 (Remarks to the Author):

This manuscript describes a quality study that combines experiment and theory. It presents new structures of terpene synthases that are useful for those who wish to understand terpene biosynthesis and make use of structures for enzyme design. The designs carried out by the authors are a meaningful step forward in this field. The insights into the non-standard DXXXXD motif are noteworthy.

That being said, there are some significant issues that should be addressed. Overall, this work suffers from some common problems encountered in biological modeling studies: lack of appreciation of the physical organic chemistry literature and speculation without support for physical effects.

Minor:

1. page 2 - Proton transfer should be added to the list of reaction steps that occur in TPSs.
2. page 10 - It is not unusual for positive charge to migrate from one side of a TPS active site to another during reaction - that is well-known.

Major:

1. It is not clear if any waters were included/allowed to be inside the active sites during simulations. Please clarify and justify.
2. A barrier of 25 kcal/mol seems too high for the biological conditions that are relevant. The authors should comment on this issue and explain why they think this barrier is okay here.
3. In many places the authors mention "proving" aspects of mechanisms. The word "prove" should be changed. They did not prove the things indicated.
4. The authors should comment explicitly on the catalytic competence of the conformations of the TPS active sites in the xray structures and the computed structures. That is a big issue in the TPS structural biology field.
5. It is dangerous to assume reactivity preferences based on distances in computed minima.
6. The authors repeatedly claim "stabilization" of carbocations. These claims should not be made without actual data such as computed relative binding energies. Proximity of sidechains to carbocations is not enough.
7. page 10: "cyclopropenyl" is not the correct term for the cation in question. That is a protonated cyclopropane. It is a type of nonclassical cation. There is an extensive literature on such structures, both relevant to terpenes and not. That literature should be discussed.

Reviewer #2 (Remarks to the Author):

The authors previously discovered the venezuelaenes and the enzyme, VenA, responsible for the cyclization, of this unique 5-5-6-7 tetracyclic diterpene core. Here, they continue their study of this interesting terpene synthase and it's an impressive amount of work. They determined the structure of VenA using Xray crystallography and use the structures to inform docking models, MD simulations, QM/MM calculations, and mutagenesis experiments to probe key residues involved in substrate selectivity, intermediate stabilization, and catalysis. Structure and mutagenesis provided key insights into the atypical Asp-rich motif, DSFVSD, and propose a conserved Gln residue that acts as a hydrogen-bonding residue to the RY diphosphate sensor. VenA was also engineered into a sesterterpene synthase by enlarging the active site with a hexamutant. They also structurally characterized four novel diterpenes including the first 5-7 bicyclic terpene of bacterial origin and a novel 6-5-6-6 tetracycle. Compound structural elucidation was done with convincing NMR and two crystal structures. This work is significant because the experimental validation of sequence-structure-function of terpene synthases will guide future engineering efforts, genome mining of new terpene synthases, and the development of biocatalysts. In addition, since the VenA enzyme was characterized and its cyclization mechanism was proposed (which begins with a 1,10-cyclization), several terpene synthases have been reported and their cyclization mechanisms studied in detail.

Therefore, this is a welcome addition to this growing family of literature on bacterial terpene synthases.

1. Overall, the writing is verbose and the text should be revised to make paragraphs/sections more concise. Relatedly, the Results section has much discussion, and the Discussion section has a lot of repeated text and ideas from the Results. This reviewer strongly suggests to revise these sections into either a combined Results and Discussion section or to revise the Results and/or Discussion sections to not be so repetitive.
2. Much is made about the “previously unknown conserved Gln residue” that takes the place of the second Asp in the DSFVSD motif and binds to Arg in the RY motif. Fig. 2d is convincing. However, Gln is not conserved. Within its closest relatives, it is Gln or Glu (and Bnd4 is Glu). So how this is described should be revised. Also, the Gln in question does not take the place of the missing second Asp to bind to Arg: the 2nd residue, Ser in VenA, Asn in Bnd4, Ser in FgGS are all bound to Arg. Were the other residues that bind to Arg in known type I terpene synthases (Fig. S6) analyzed? Perhaps there is another residue that also binds Arg and this “conserved Gln” is just a functional replacement of those residues?
3. A few related references are missing:
 - a. Bnd4 is discussed and used for sequence analysis. One recent paper on mutagenesis of Bnd4 was not cited or discussed (10.1039/D2OB01931K). In fact, several of the same residues were mutated in that study, some with different results (soluble/active vs insoluble mutants) and some similar results (formation of GGOH, GLOH, springenes, and cembrenes). It may be of note that F185A gave the venezuelaene alcohols while Bnd4 F162A gave GGOH/GLOH.
 - b. A recent paper on another terpene synthase, AlbS, that catalyzes initial 1,10-cyclization, also with mutagenesis and mechanistic studies (10.1016/j.chempr.2022.12.006). As all other initial 1,10-cyclization enzymes are compared here, and it is frequently highlighted that this is the first enzyme with initial 1,10-cyclization to be structurally characterized, it would make sense to add AlbS to these analyses (i.e., Table S12, Fig S87)
4. It is stated that retention indices were used to compare products with authentic standards from the literature and NIST, but only retention times are given. In addition, none of the new compounds have retention indices given. A retention time is not very useful for comparison in the community. Please calculate and provide retention indices, most importantly for the new compounds.
5. Fig 5: The cations and R/S labels are very small. Some of the potential interactions look very far away and even appear to be going through other atoms in these orientations. Such as Y88 to C1 in inset C and F215 to C15 in panel D. Can any other orientations better show? What are the distances?

Ln 35. The term “conservative” is used throughout, this should be “conserved”

Ln 36: The description of conserved glutamine and “another hydrogen bond forming amino acid” is not very descriptive and perhaps not important enough for the abstract. Also, add hyphen between “bond forming”.

Ln 51, 239: “linear” is better described as “acyclic” considering isoprenoid precursors are branched.

Ln 87 (and elsewhere): “Markovnikov additive 1,10-cyclization” sounds awkward. Perhaps just “Markovnikov 1,10-cyclization”?

Ln 166 (and elsewhere): The terms “terpenes” and “terpenoids” are used throughout. It’s not clear if these are supposed to be interchangeable or if the authors are using them distinctively.

Ln 178: The use of “we revealed” is awkward after a literature search. Most literature on terpene synthases, particularly recent studies, structural or otherwise, state this.

Ln 185: Wording here is confusing about plant versus bacterial TSs. Is this statement saying that only the 3rd Asp in plant enzymes coordinate with Mg²⁺? This is true for selinadiene synthase. Perhaps just a text revision will clarify here.

Ln 210: B1/B2 “kinker”. Would “the B1/B2 helix-break” to match “G1/G2 helix-break” be better?

Ln 295 (and elsewhere): (Ha) and PPI should have subscript “a” and “i”

Ln 482 and Table S11: SvS-A2 is a reconstructed ancestor (unlikely to be the exact natural ancestor). Thus, it is a bit strange to say this enzyme uses GGPP as a “natural” substrate. Perhaps you could mention it is a reconstructed ancestor of SvS?

Ln 257: Is “spontaneity” the correct term here? Spontaneous does not equal fast, and the enzyme is controlling the 1,10-cyclization.

Ln 314: Revise grammar: "cation is destructed as C2 removing from C7 and approaching to C6"
Ln 343: Do not use "synthetic" when describing an enzyme mechanism
Ln 352: How was the titer of dolabellatriene calculated in this study? It is not clear from the methods.
Ln 274: "GGPP is the more favorable substrate of VenA." The energy barrier for 1,10-cyclization is not the only consideration here, the binding of the different length prenyl-PPs will also influence substrate preference.
Ln 405: These conclusions are known/predicted/assumed about all terpene synthases. Perhaps it could be concisely revised, and stated that this follows that of other terpene synthases.
Ln 532: "DTS with higher preference for GGPP (VenAF185A) than VenAWT." Higher preference for GGPP over other prenyl chain lengths. Maybe "selectivity" would be a better word choice here?
Ln 558: Water quenching a carbocation is not "activation of an inert C-H bond by regioselective incorporation of a water molecule.

Tables S2, S11: reformatting would make these easier to read

Table S5–S8: the single- and double-headed blue arrows for HMBC and NOESY are too difficult to tell apart. Perhaps change the color of the NOESY arrows.

Table S9: left align the cell for fusicoccadiene synthase

Fig S5: Y88A deprotonation is missing a "-H+"

Fig S7: It's not clear why only some seemingly unrelated terpene synthases are annotated on this network. Are any of the other enzymes discussed in this paper on this network? And if not, why were they not found in UniProt, since the phylogenetic tree puts several enzymes, such as Bnd4, close to VenA. Also, this figure is referenced at Ln 215 in discussion with Bnd4, but this is unrelated to Bnd4.

Fig S8: this is not a complete sequence alignment, only 150 residues.

Fig S44, S73 legends: "lablled" typo

Fig S87: Nephthenol are drawn with the incorrect configuration

Fig S88 and 1b: "Cemebrene" typo

Reviewer #3 (Remarks to the Author):

This manuscript describes the structure and engineering of diterpene synthase VenA. The structure of VenA were solved in its apo and holo-forms with magnesium ions and diphosphates without substrate bound. Binding and conformation of the substrate were analyzed by docking and MD simulations. Mutagenesis experiments explained the involvement of aromatic residues in cation-mediated cyclization and generating new diterpenoids. The manuscript contains solid structural data and is well written. One of the key findings of this manuscript is that VenA has an atypical "Asp-rich" motif. However, since in several other enzymes, the second and third Asp of the canonical motif are less important for catalysis, so it is not very surprising that VenA lacks the third Asp residue, and the second Asp is not essential. It is also well known that aromatic amino acids are important to stabilize carbocation intermediates, as many similar structural studies of terpene cyclases have been reported. The production of new terpenoids using VenA mutants is interesting. However, mutation of aromatic residues to interrupt the natural cation-mediated cyclization mechanism is a common strategy to produce unnatural terpenoids, and has been reported in many previous studies. Therefore, I do not believe that this manuscript represents novelty suitable for publication in this journal.

Reviewer 1's comments to authors: (Authors' responses to reviewers are in blue)

This manuscript describes a quality study that combines experiment and theory. It presents new structures of terpene synthases that are useful for those who wish to understand terpene biosynthesis and make use of structures for enzyme design. The designs carried out by the authors are a meaningful step forward in this field. The insights into the non-standard DXXXXD motif are noteworthy.

We appreciate the positive comments from this reviewer.

That being said, there are some significant issues that should be addressed. Overall, this work suffers from some common problems encountered in biological modeling studies: lack of appreciation of the physical organic chemistry literature and speculation without support for physical effects.

We appreciate the constructive criticisms and suggestions for helping us improve our work. As suggested, a number of important references (refs. 7, 8 and 51-55) have been cited (lines 58 and 320).

Minor:

1. page 2 - Proton transfer should be added to the list of reaction steps that occur in TPSs.

Added as suggested (line 57).

2. page 10 - It is not unusual for positive charge to migrate from one side of a TPS active site to another during reaction - that is well-known.

We have deleted this sentence (lines 340-341).

Major:

1. It is not clear if any waters were included/allowed to be inside the active sites during simulations. Please clarify and justify.

We appreciate this important concern. To be concise and show the binding poses more clearly, we chose not to show water molecules in the figures of enzyme-ligand

complexes (Figs. 4 and 6) in the previous version. Now we realize that it may confuse readers. Indeed, we considered the behavior of water molecules and did not apply any restriction on them during the whole MD simulations. Since we found that water molecules did not exhibit any notable effect on either substrates binding or enzymatic catalysis during simulations, we did not discuss it in our previous manuscript. In the revised version, we have added “The water molecules are hidden to better present the zoom-in pocket (see Supplementary Fig. 16 for details)” to the legends of Figs. 4 and 6 to avoid any potential confusions from readers about the absence of water molecules. Meanwhile, we have supplemented the following figure in Supplementary Information (Supplementary Fig. 16) to present more details of VenA’s active pocket in different models, in which the water molecules are explicitly shown.

Note: The QM/MM models and zoom-in active sites of VenA with GPP, FPP or GGPP

binding and of VenA^{Y88A/W107A/V111A/T112A/F215A/F219A} with GFP binding. The water molecules around the active pocket of wild-type or mutant VenA are shown.

2. A barrier of 25 kcal/mol seems too high for the biological conditions that are relevant. The authors should comment on this issue and explain why they think this barrier is okay here.

Thanks for the very constructive feedback. A relative barrier of 25 kcal/mol predicted in this study is reasonable as we take the following reasons into account: (1) In view of our previous computational simulations, the PP_i cleavage step (or the concerted cleavage & cyclization process) is often the rate-determining step with a free energy barrier of more than 20 kcal/mol.^{1, 2, 3} To conserve computational resources, sometime we performed potential energy calculations instead of free energy calculations.^{4, 5} Generally, the relative reaction barrier is almost similar or a little bit higher than the free energy barrier in several case studies of TPSs. (2) In this study, the major objective of the relative reaction energy calculations is to compare the barrier difference among different systems, rather than their absolute values. Thus, we did not perform the very expensive reaction free energy simulations. If we take the entropy effect into consideration, we deduce that the barrier (ΔG^\ddagger) will decrease while the $\Delta\Delta G^\ddagger$ value likely be the same as observed in our current calculations.

3. In many places the authors mention "proving" aspects of mechanisms. The word "prove" should be changed. They did not prove the things indicated.

We have changed the inappropriately used "prove" words accordingly. Please see lines 79, 257 and 264.

4. The authors should comment explicitly on the catalytic competence of the conformations of the TPS active sites in the xray structures and the computed structures. That is a big issue in the TPS structural biology field.

We agree with the reviewer that the conformational changes of the active-site residues

during the simulation process are a crucial issue, especially when compared to the crystal structure. To address this issue, we have added a new figure (Supplementary Fig. 26, *please see below*) and some related descriptions for these changes (lines 344-346).

Supplementary Figure 26. Superimposition of the active pockets of the X-ray structure and multiscale simulation models: substrate (a) and product (b) states. Residues from the crystal structure and simulation models are shown in grey and cyan, respectively. The RMSD values are 1.080 Å and 1.149 Å, respectively.

5. It is dangerous to assume reactivity preferences based on distances in computed minima.

We appreciate this comment regarding the limitations of simply using distance evaluation to infer reaction preferences. We should point out that we not only provided distances but also calculated the potential energy curves of different cyclization modes to determine their preferences in our simulations of the wild type VenA. While for the mutated enzyme that can use GFPP as a substrate, we inferred preferences solely based on distances as it is reasonable in many cases. The conformation captured from the QM/MM optimization is a near attack conformation (NAC). One can foretell that when the C–O bond breaks, the carbocation formed at C1 would be well prepared to attack the C14=C15 double bond (as shown in the diagram below, residues are omitted for clarity). However, the secondary carbocation derived from the 1-15 cyclization would be unstable while the 1-14 cyclization is more favorable since the resulted intermediate would be a tertiary carbocation, which is in good agreement with the experimental

result. Besides, since the focus of this study is not the complete catalytic mechanism of the mutated VenA on GFPP, we did not discuss the possibilities of different cyclization modes in terms of energy in order for saving computing resource. Therefore, we made the preference inferences based on optimized structures. Actually, the NAC or pre-reaction state concepts have been used in many previous studies^{6, 7, 8, 9}. We agree with the reviewer that this method may not always work and acknowledge that further exploration of different cyclization modes would be valuable and will be discussed in our future study of the whole catalytic process in a due course.

6. The authors repeatedly claim "stabilization" of carbocations. These claims should not be made without actual data such as computed relative binding energies. Proximity of sidechains to carbocations is not enough.

We appreciate the constructive comments. In the revised manuscript, we have addressed this concern by computing the relative binding energies between key residues and carbocations (please see the newly added Supplementary Table 3 and Supplementary Method of "Binding energy calculation"). This approach provides more precise assessment of stabilization and further reinforces our findings on the crucial role of the key amino acid residues involved in the VenA catalysis. The related descriptions have been revised accordingly (lines 306-309 and 404-405).

7. page 10: "cyclopropenyl" is not the correct term for the cation in question. That is a protonated cyclopropane. It is a type of nonclassical cation. There is an extensive literature on such structures, both relevant to terpenes and not. That literature should be

discussed.

In the revised manuscript, we have corrected the incorrect terminology to “protonated cyclopropane cation” as widely used in other works and provided some brief discussion on this “nonclassical carbocation”. Please see lines 319-322, 326 and 335-336 for details.

Reviewer 2’s comments to authors: (Authors’ responses to reviewers are in red)

The authors previously discovered the venezuelaenes and the enzyme, VenA, responsible for the cyclization, of this unique 5-5-6-7 tetracyclic diterpene core. Here, they continue their study of this interesting terpene synthase and it’s an impressive amount of work. They determined the structure of VenA using Xray crystallography and use the structures to inform docking models, MD simulations, QM/MM calculations, and mutagenesis experiments to probe key residues involved in substrate selectivity, intermediate stabilization, and catalysis. Structure and mutagenesis provided key insights into the atypical Asp-rich motif, DSFVSD, and propose a conserved Gln residue that acts as a hydrogen-bonding residue to the RY diphosphate sensor. VenA was also engineered into a sesterterpene synthase by enlarging the active site with a hexamutant. They also structurally characterized four novel diterpenes including the first 5-7 bicyclic terpene of bacterial origin and a novel 6-5-6-6 tetracycle. Compound structural elucidation was done with convincing NMR and two crystal structures. This work is significant because the experimental validation of sequence-structure-function of terpene synthases will guide future engineering efforts, genome mining of new terpene synthases, and the development of biocatalysts. In addition, since the VenA enzyme was characterized and its cyclization mechanism was proposed (which begins with a 1,10-cyclization), several terpene synthases have been reported and their cyclization mechanisms studied in detail. Therefore, this is a welcome addition to this growing family of literature on bacterial terpene synthases.

We highly appreciate the positive and encouraging comments from this reviewer.

1. Overall, the writing is verbose and the text should be revised to make

paragraphs/sections more concise. Relatedly, the Results section has much discussion, and the Discussion section has a lot of repeated text and ideas from the Results. This reviewer strongly suggests to revise these sections into either a combined Results and Discussion section or to revise the Results and/or Discussion sections to not be so repetitive.

We thank the reviewer for his/her instructive suggestions for helping us improve the quality of our work. Following the reviewer's suggestions, we have revised the Results and Discussion sections by deleting/revising the repetitive descriptions (lines 233-235, 424-427, 507-513, 519-522, 538-541, and 583-585).

2. Much is made about the “previously unknown conserved Gln residue” that takes the place of the second Asp in the DSFVSD motif and binds to Arg in the RY motif. Fig. 2d is convincing. However, Gln is not conserved. Within its closest relatives, it is Gln or Glu (and Bnd4 is Glu). So how this is described should be revised. Also, the Gln in question does not take the place of the missing second Asp to bind to Arg: the 2nd residue, Ser in VenA, Asn in Bnd4, Ser in FgGS are all bound to Arg. Were the other residues that bind to Arg in known type I terpene synthases (Fig. S6) analyzed? Perhaps there is another residue that also binds Arg and this “conserved Gln” is just a functional replacement of those residues?

We highly appreciate these important comments and suggestions. As pointed out by this reviewer, the Gln residue is not conserved as shown in Fig. 2d. Thus, the above-mentioned sentence in Abstract has been revised to avoid potential misunderstanding as follow: “Functional and structural investigations on the atypical ¹¹⁵DSFVSD¹²⁰ motif of VenA (*versus* the canonical “DDXX(X)D/E” motif) reveal that the absent second Asp of canonical motif is functionally replaced by Ser116 and a previously unknown Gln83 to form a hydrogen bond network with Arg344, together with bioinformatics analysis suggesting a new subclass of type I microbial TPSs” (lines 34-39). According to our re-analysis of the reported crystal structures of bacterial type I TPSs shown in Supplementary Fig. 6, no other residues that bind to the conserved Arg are found in the crystal structures of CotB2 and 1,8-cineole synthase (*see below*). Thanks to this

question, we found that Gln50 and Tyr69 (previously missing from our analysis, *see below* and revised Supplementary Fig. 6) form hydrogen bonds with the conserved Arg of selinadiene synthase and EIZS, respectively. Also, it could be speculated that the residues with similar functions of “conserved Q/E” might exist, but unrevealed in type I TPSs containing the atypical Asp-rich motif. With no doubt, it is worth further explorations from this work.

Note: All the residues around 4 Å of the conserved Arg are shown and the distances (Å) between different atoms are shown and the distances < 3.4 Å are considered for the formation of hydrogen bonds. Only the closed forms bound with trinuclear magnesium cluster are selected. Mg²⁺ ions are marked by green spheres; the salt bridge, coordination and hydrogen bonds are marked by orange, blue and black dashed lines, respectively.

3. A few related references are missing:

a. Bnd4 is discussed and used for sequence analysis. One recent paper on mutagenesis of Bnd4 was not cited or discussed (10.1039/D2OB01931K). In fact, several of the

same residues were mutated in that study, some with different results (soluble/active vs insoluble mutants) and some similar results (formation of GGOH, GLOH, springenes, and cembrenes). It may be of note that F185A gave the venezuelaene alcohols while Bnd4 F162A gave GGOH/GLOH.

We apologize for missing this important and interesting work of Bnd4. In the revised manuscript, this paper has been cited and discussed (lines 592-597).

b. A recent paper on another terpene synthase, AlbS, that catalyzes initial 1,10-cyclization, also with mutagenesis and mechanistic studies (10.1016/j.chempr.2022.12.006). As all other initial 1,10-cyclization enzymes are compared here, and it is frequently highlighted that this is the first enzyme with initial 1,10-cyclization to be structurally characterized, it would make sense to add AlbS to these analyses (i.e., Table S12, Fig S87)

It is a beautiful work in the field of bacterial DTSs. Following the suggestion, we have included AlbS in our bioinformatics analyses (please see the revised Supplementary Table 13 and Supplementary Fig. 88) and the reference has been cited (line 491).

4. It is stated that retention indices were used to compare products with authentic standards from the literature and NIST, but only retention times are given. In addition, none of the new compounds have retention indices given. A retention time is not very useful for comparison in the community. Please calculate and provide retention indices, most importantly for the new compounds.

Per this important suggestion, the retention indices of 7 (1917) and 18* (1525) comparing with those of the authentic standards from the literature and NIST have been given the part of “Structure determination of 4, 5, 7, 13, 14, 15, 16, 17 and 18*” in Supplementary Information. Also, the retention indices of new compounds have been calculated and listed in the part of “Structure elucidation of 8, 9, 10, 11, 12 and 19” in Supplementary Information.

5. Fig 5: The cations and R/S labels are very small. Some of the potential interactions

look very far away and even appear to be going through other atoms in these orientations. Such as Y88 to C1 in inset C and F215 to C15 in panel D. Can any other orientations better show? What are the distances?

We have revised Figure 5 to better display the details of this important figure. Based on the dynamics of the active pocket of VenA and reaction intermediates, together with the informative product structures resulted from mutagenesis of the active-site residues, we propose and illustrate the putative interactions between the carbon cations and the specific amino acids within the range of 6.5 Å (lines 788-789). Moreover, as suggested by reviewer #1, we performed additional calculations of relative binding energies to consolidate the assignment of these potential cation- π interactions (please see the newly added Supplementary Table 3).

Ln 35. The term “conservative” is used throughout, this should be “conserved”

Revised throughout the revised manuscript (lines 141, 198, 201, 232, 518 and 526).

Ln 36: The description of conserved glutamine and “another hydrogen bond forming amino acid” is not very descriptive and perhaps not important enough for the abstract. Also, add hyphen between “bond forming”.

The sentence has been revised as follow: “Functional and structural investigations on the atypical ¹¹⁵DSFVSD¹²⁰ motif of VenA (*versus* the canonical “DDXX(X)D/E” motif) reveal that the absent second Asp of canonical motif is functionally replaced by Ser116 and a previously unknown Gln83 to form a hydrogen bond network with Arg344, together with bioinformatics analysis suggesting a new subclass of type I microbial TPSs” (lines 34-38).

Ln 51, 239: “linear” is better described as “acyclic” considering isoprenoid precursors are branched.

Revised (lines 54 and 246).

Ln 87 (and elsewhere): “Markovnikov additive 1,10-cyclization” sounds awkward.

Perhaps just “Markovnikov 1,10-cyclization”?

Revised (lines 91, 256 and 375).

Ln 166 (and elsewhere): The terms “terpenes” and “terpenoids” are used throughout. It’s not clear if these are supposed to be interchangeable or if the authors are using them distinctively.

Terpenes and terpenoids are not interchangeable. To state these two terms more accurately, we have made changes in lines 50, 101 and 294-295.

Ln 178: The use of “we revealed” is awkward after a literature search. Most literature on terpene synthases, particularly recent studies, structural or otherwise, state this.

We revised this sentence as follow: “The literature research and investigation on the Asp-rich motifs of representative crystal structures indicate that...” (lines 181-182).

Ln 185: Wording here is confusing about plant versus bacterial TSs. Is this statement saying that only the 3rd Asp in plant enzymes coordinate with Mg^{2+} ? This is true for selinadiene synthase. Perhaps just a text revision will clarify here.

We have rephrased the statement as follow: “The third Asp/Glu is usually located outside of the catalytic pocket and hence dispensable for the catalytic activity, or in some cases coordinated with Mg^{2+} as selinadiene synthase” (lines 189-190).

Ln 210: B1/B2 “kinker”. Would “the B1/B2 helix-break” to match “G1/G2 helix-break” be better?

We have changed “B1/B2 kinker” into “B1/B2 helix-break” (line 215).

Ln 295 (and elsewhere): (H_a) and PP_i should have subscript “a” and “i”

Per this suggestion, “a” and “i” of H_a and PP_i have been subscripted throughout the revised manuscript and Supplementary Information.

Ln 482 and Table S11: SvS-A2 is a reconstructed ancestor (unlikely to be the exact

natural ancestor). Thus, it is a bit strange to say this enzyme uses GGPP as a “natural” substrate. Perhaps you could mention it is a reconstructed ancestor of SvS?

We appreciate this point and have revised the related parts accordingly (lines 501-502 and the revised Supplementary Table 12).

Ln 257: Is “spontaneity” the correct term here? Spontaneous does not equal fast, and the enzyme is controlling the 1,10-cyclization.

We have replaced “spontaneity” with “preference” (line 265) in the revised manuscript.

Ln 314: Revise grammar: “cation is destructed as C2 removing from C7 and approaching to C6”

We thank the reviewer for bringing this grammar error to our attention. In the new version, we have revised the sentence to “Subsequently, the structure of C2-C6-C7 protonated cyclopropane cation is disrupted, with C2 separating from C7 and approaching C6 to form a regular C-C single bond” (lines 327-329).

Ln 343: Do not use “synthetic” when describing an enzyme mechanism

We have revised “synthetic” to “biosynthetic” (lines 359 and 798).

Ln 352: How was the titer of dolabellatriene calculated in this study? It is not clear from the methods.

To address this concern, we have added the quantification method in the part of “Isolation, purification, and characterization of **8**, **9**, **10**, **11**, **12** and **19**” in Supplementary Methods: “Then, the supernatant was directly used for GC/GC-MS analysis to monitor the terpene/terpenoid production. The yield of **9** was calculated by comparing the peak areas of two independent fermentation samples with that of an authentic standard with known concentration during GC analysis”.

Ln 274: “GGPP is the more favorable substrate of VenA.” The energy barrier for 1,10-cyclization is not the only consideration here, the binding of the different length prenyl-

PPs will also influence substrate preference.

To address this concern, we have deleted this problematic statement (lines 281-282).

Ln 405: These conclusions are known/predicted/assumed about all terpene synthases. Perhaps it could be concisely revised, and stated that this follows that of other terpene synthases.

We have deleted the whole paragraph of these well-known information, also avoiding repetitive descriptions in the sections of Results and Discussion (424-427).

Ln 532: “DTS with higher preference for GGPP (VenA^{F185A}) than VenA^{WT}.” Higher preference for GGPP over other prenyl chain lengths. Maybe “selectivity” would be a better word choice here?

We have changed “preference” into “selectivity” as suggested (line 557).

Ln 558: Water quenching a carbocation is not “activation of an inert C-H bond by regioselective incorporation of a water molecule.

We appreciate the reviewer for pointing out this mistake. This wrong statement has been removed in the revised manuscript (line 583-585).

Tables S2, S11: reformatting would make these easier to read

Reformatted.

Table S5–S8: the single- and double-headed blue arrows for HMBC and NOESY are too difficult to tell apart. Perhaps change the color of the NOESY arrows.

Color changed. Please see the revised Supplementary Tables 6-9.

Table S9: left align the cell for fusicoccadiene synthase

The format has been refined. Please see the revised Supplementary Table 10.

Fig S5: Y88A deprotonation is missing a “-H⁺”

We appreciate the careful reading here. We have added “-H⁺” into the revised Supplementary Fig. 5.

Fig S7: It's not clear why only some seemingly unrelated terpene synthases are annotated on this network. Are any of the other enzymes discussed in this paper on this network? And if not, why were they not found in UniProt, since the phylogenetic tree puts several enzymes, such as Bnd4, close to VenA. Also, this figure is referenced at Ln 215 in discussion with Bnd4, but this is unrelated to Bnd4.

We highly appreciate these comments. We double checked the SSN network and found that the functions of unrelated terpene synthases in Supplementary Fig. 7 were predicted by UniProt database without experimental evidences. To avoid unnecessary misunderstanding, the names of unrelated terpene synthases are removed. This SSN network is only used to find VenA homologues deposited in UniProt database by using the sequence of VenA as a probe (the obtained sequences show $\geq 49\%$ identity with VenA). We also revised the related description as follow: “Based on sequence similarity network (SSN)-based homologues search, sequence alignment analysis of VenA, Bnd4 and VenA homologues collected from UniProt database, and AlphaFold2 predicted structures (Supplementary Figs. 7-10) ...” (Lines 218-221).

Fig S8: this is not a complete sequence alignment, only 150 residues.

To address this issue, the complete sequence alignment has been included in the revised Supplementary Fig. 8.

Fig S44, S73 legends: “lablled” typo

The typos have been corrected in the figure legends of Supplementary Figs. 45 and 74.

Fig S87: Nephthenol are drawn with the incorrect configuration

The structure of (*R*)-nephthenol has been redrawn with the correct configuration. Please

see the new Supplementary Fig. 88.

Fig S88 and 1b: “Cemebrene” typo

We have corrected “cemebrene” to “cembrene” in the revised Fig. 1 and Supplementary Fig. 89.

Reviewer 3’s comments to authors: (Authors’ responses to reviewers are in Blue)

This manuscript describes the structure and engineering of diterpene synthase VenA. The structure of VenA were solved in its apo and holo-forms with magnesium ions and diphosphates without substrate bound. Binding and conformation of the substrate were analyzed by docking and MD simulations. Mutagenesis experiments explained the involvement of aromatic residues in cation-mediated cyclization and generating new diterpenoids. The manuscript contains solid structural data and is well written.

We appreciate the positive comments from this reviewer.

One of the key findings of this manuscript is that VenA has an atypical “Asp-rich” motif. However, since in several other enzymes, the second and third Asp of the canonical motif are less important for catalysis, so it is not very surprising that VenA lacks the third Asp residue, and the second Asp is not essential.

We respectfully disagree with these comments. First, “VenA has an atypical ‘Asp-rich’ motif” is not the key finding of this work, which has already been reported in our previous paper (*ACS Catal.* 2020, 10, 5846-5851). The key findings of this work include: 1) the first crystal structures of a bacterial type I diterpene synthase with an atypical “Asp-rich” motif (DXXXXD), as well as the first structural basis for the initial 1,10-cyclization mode of diterpene synthases; 2) the finding that the absent second Asp of the canonical “DDXX(X)D/E” motif is functionally replaced by Ser116 and a previously unknown Gln83 to form a hydrogen bond network with Arg344, together with bioinformatics analysis suggesting a new subclass of type I microbial TPSs; 3) the catalytic mechanism of VenA based on systematic mutagenesis analysis and multiscale computational simulations; and 4) the structure-guided enzyme engineering leading to

a group of new terpenoid products, especially a bacterial-derived sesterterpene. These novelties are well acknowledged by Reviewer 1 and 2 (also partially acknowledged by Reviewer 3, *see above*). Second, the second and third Asp residues in a typical “Asp-rich” motif are less important for catalysis in some cases, but in some other cases (*e.g.*, CotB2, EIZS and pentalenene synthase) the second Asp is catalytically essential (lines 186-188). Third, the atypical “Asp-rich” motif of VenA (¹¹⁵DSFVSD¹²⁰) does not “lack the third Asp residue” as raised by this reviewer.

It is also well known that aromatic amino acids are important to stabilize carbocation intermediates, as many similar structural studies of terpene cyclases have been reported. The production of new terpenoids using VenA mutants is interesting. However, mutation of aromatic residues to interrupt the natural cation-mediated cyclization mechanism is a common strategy to produce unnatural terpenoids, and has been reported in many previous studies. Therefore, I do not believe that this manuscript represents novelty suitable for publication in this journal.

We have used the “common” strategy, together with our new structure-guided enzyme engineering, to generate “uncommon” outcome: four new diterpenoids including a novel 6-5-6-6 skeleton and a sesterterpene resulted from a semi-rationally engineered diterpene synthase. It is also worth noting that the structure-guided enzyme engineering is only a small portion of novelties of this work.

References:

1. Zhang, F., Chen, N., Zhou, J., Wu, R. Protonation-dependent diphosphate cleavage in FPP cyclases and synthases. *ACS Catal.* **6**, 6918-6929 (2016).
2. Zhang, F., Wang, Y.-H., Tang, X., Wu, R. Catalytic promiscuity of the non-native FPP substrate in the TEAS enzyme: non-negligible flexibility of the carbocation intermediate. *Phys. Chem. Chem. Phys.* **20**, 15061-15073 (2018).
3. Zhuang, J., *et al.* Insights into the enzymatic catalytic mechanism of bCinS: the importance of protein conformational change. *Catal. Sci. Technol.* **12**, 1651-1662 (2022).
4. Wang, Y.-H., *et al.* Catalytic role of carbonyl oxygens and water in selinadiene synthase.

Nat. Catal. **5**, 128-135 (2022).

5. Zha, W., *et al.* Rationally engineering santalene synthase to readjust the component ratio of sandalwood oil. *Nat. Commun.* **13**, 2508 (2022).
6. Hur, S., Bruice, T. C. Enzymes do what is expected (chalcone isomerase versus chorismate mutase). *J. Am. Chem. Soc.* **125**, 1472-1473 (2003).
7. Hur, S., Bruice, T. C. The near attack conformation approach to the study of the chorismate to prephenate reaction. *Proc. Natl. Acad. Sci. USA* **100**, 12015-12020 (2003).
8. Hur, S., Bruice, T. C. The mechanism of catalysis of the chorismate to prephenate reaction by the *Escherichia coli* mutase enzyme. *Proc. Natl. Acad. Sci. USA* **99**, 1176-1181 (2002).
9. Schaus, L., *et al.* Protoglobin-catalyzed formation of *cis*-trifluoromethyl-substituted cyclopropanes by carbene transfer. *Angew. Chem. Int. Ed.* **135**, e202208936 (2023).

REVIEWERS' COMMENTS

Reviewer #1 (Remarks to the Author):

I appreciate the authors' efforts to address all reviewer comments and believe this work should now be published.

One small comment: ref 52 is not particularly relevant and can be deleted. The other references added should suffice, but if the authors would like to add a terpene-relevant non-classical ion paper, this one is better: Hong, Y. J.; Tantillo, D. J. Chem. Sci. 2013, 4, 2512, C-H-pi Interactions as Modulators of Carbocation Structure. Implications for Terpene Biosynthesis

Reviewer #2 (Remarks to the Author):

The authors adequately addressed this reviewer's concerns. Thank you.

Reviewer 1's comments to authors: (Authors' responses to reviewers are in blue)

I appreciate the authors' efforts to address all reviewer comments and believe this work should now be published.

We appreciate the final approval of our work by this reviewer.

One small comment: ref 52 is not particularly relevant and can be deleted. The other references added should suffice, but if the authors would like to add a terpene-relevant non-classical ion paper, this one is better: Hong, Y. J.; Tantillo, D. J. Chem. Sci. 2013, 4, 2512, C-H- π Interactions as Modulators of Carbocation Structure. Implications for Terpene Biosynthesis

The previous ref. 52 has been replaced by the suggested reference in the revised manuscript.

Reviewer 2's comments to authors:

The authors adequately addressed this reviewer's concerns. Thank you.

We appreciate the reviewer's kind help for improving our work. Thanks a lot.